# Unlocking osmotic energy harvesting potential in challenging real-world hypersaline environments through vermiculite-based hetero-nanochannels

Jin Wang [1] ✉, Zheng Cui[1], Shangzhen Li[1], Zeyuan Song[1], Miaolu He[1], Danxi Huang[1], Yuan Feng[1], YanZheng Liu[1], Ke Zhou [2] ✉, Xudong Wang[1] & Lei Wang [1] ✉

Nanochannel membranes have demonstrated remarkable potential for osmotic energy harvesting; however, their efficiency in practical high-salinity systems is hindered by reduced ion selectivity. Here, we propose a dual-separation transport strategy by constructing a two-dimensional (2D) vermiculite (VMT)-based heterogeneous nanofluidic system via an eco-friendly and scalable method. The cations are initially separated and enriched in micropores of substrates during the transmembrane diffusion, followed by secondary precise sieving in ultra-thin VMT laminates with high ion flux. Resultantly, our nanofluidic system demonstrates efficient osmotic energy harvesting performance, especially in hypersaline environment. Notably, we achieve a maximum power density of 33.76 W m$^{-2}$, a 6.2-fold improvement with a ten-fold increase in salinity gradient, surpassing state-of-the-art nanochannel membranes under challenging conditions. Additionally, we confirm practical hypersaline osmotic power generation using various natural salt-lake brines, achieving a power density of 25.9 W m$^{-2}$. This work triggers the hopes for practical blue energy conversion using advanced nanoarchitecture.

The surging global energy requirements have stimulated a burgeoning fascination in development of clean energy[1]. Osmotic energy, an eco-friendly and renewable blue energy acquired through the from the mixing of waters with varying salinities has gained widespread interest for its potential for mitigating the adverse impacts of traditional energy generation methods on the environment[2,3]. Based on theoretical calculations the osmotic energy only contained in oceans is estimated to reach approximately 2.6 terawatts (TW)[4]. Additionally, vast reserves of salinity gradient energy are also present in other potential sources, including anthropogenic sources such as mining wastewater and seawater desalination brine, and natural sources such as salt lakes brine and brackish water[5]. Among the strategies developed for osmotic energy harvesting, membrane-based reverse electrodialysis (RED) has emerged as the most promising method to date. In the RED process, the Gibbs free energy released during mixing of solutions with different concentrations can directly be converted into electricity through the selective transport of ions with specific charge polarity[3,6]. The advances in nanotechnology obtained in the past two decades also greatly encouraged the development of ionic energy conversion. 2D Nanofluidic channel systems with controllable ion transport have

[1]Research Institute of Membrane Separation Technology of Shaanxi Province, Key Laboratory of Membrane Separation of Shaanxi Province, School of Environmental & Municipal Engineering, Xi'an University of Architecture and Technology, Xi'an, China. [2]College of Energy, Soochow Institute for Energy and Materials InnovationS (SIEMIS), Jiangsu Provincial Key Laboratory for Advanced Carbon Materials and Wearable Energy Technologies, Soochow University, Suzhou, China. ✉e-mail: wangjin@xauat.edu.cn; zhouke@suda.edu.cn; wl0178@126.com

emerged as a novel approach for osmotic energy harvesting[7–9]. For instance, a single-layer $MoS_2$ has achieved a power density of approximately $10^6\,W\,m^{-2}$, which is a million times higher than that attained by traditional exchange membranes[6]. Although these results certainly provide a new source of inspiration for the development of osmotic power harvesting technology, there are many challenges currently still impede the industrial applications of 2D nanochannel membranes. For instance, the exfoliation of most 2D nanomaterials typically requires the use of harsh chemicals such as strong acids, oxidants, or solvents to break the out-of-plane bonds in the layered structure[10,11]. In addition, the process of synthesizing precursors, such as MAX phase for MXene, that do not occur naturally, can be intricate[12]. VMT is a cost-effective naturally occurring clay mineral with a global production of approximately 500,000 tons[13]. Because $Si^{4+}$ and $Al^{3+}$ are uniformly substituted with low-valent ions, VMT layers naturally exhibit negative charge property. These charges can be balanced by the adsorption of cations, such as $Mg^{2+}$ and $K^+$, between the layers. Preliminary investigations have demonstrated that bulk VMT can be readily exfoliated in aqueous solution to obtain two-dimensional nanosheets with atomic-level thickness by the more environmentally friendly and gentler ion-exchange method[14,15].

Recent year, the development of various nanochannel design strategies aimed at enhancing osmotic energy conversion efficiency have been seen[16,17]. Particularly, inspired by the high-efficient ion transport observed in biological ion channels, many methodologies have been proposed for the fabrication of bionic artificial nanochannels. For instance, biomimic membranes with asymmetric nanoporous structures or surface charges exhibit unique one-way ionic transport properties, preventing power dissipation in the form of Joule heat, thus facilitating osmotic power generation[18,19]. However, the complex production processes of nanochannel membranes make them difficult to customize for full-scale operation, leading to limited industrial production[20,21]. Moreover, nanochannel membranes generally exhibit decreased selectivity and suboptimal energy harvesting performance in practical high-salinity environments due to the reduced spatial range of the electrostatic effect of the electric double layer (EDL) and charge screening effect caused by high concentrations of counterions[2]. Despite the extensive research on osmotic energy conversion from mixing river and seawater, the practical application is still limited to laboratory scale under ideal conditions. Other high-salinity resources, such as industrial wastewater and natural salt lakes, remain largely untapped[22]. Consequently, there is an urgent need for new nanostructure design concepts to facilitate the further development of high-performance membrane with facile preparation process, especially for applications in practical high-salinity environments.

In this work, we propose a dual-separation transport mechanism based on the heterogeneous nanochannel for osmotic energy harvest. The atomic-thick VMT nanosheets with large lateral size (~12 μm) are obtained via simple eco-friendly liquid exfoliation method, asymmetric nanoarchitecture is then constructed by stacking them onto a porous polyvinylidene fluoride (PVDF) substrate. Based on the theoretical simulation, the pre-storage and -separation of charged micropores reconfigure the system of salinity differences within the nanochannel, while the ultrathin VMT laminate ensure high ion selectivity and ion flux, ultimately enabling extraordinary osmotic energy harvest performance. The heterogeneous membrane with only 30 nm thickness of the VMT layer exhibit a stable high-power density of $5.45\,W\,m^{-2}$, which has exceeded the commercial benchmarking in artificial seawater/river system. Moreover, the maximum power density improved by 6.2 times with a 10-fold increase in salinity gradient, superior to state-of-the-art nanochannel membranes. When using the natural salt-lake brine to explore the application potential, a maximum value of $25.9\,W\,m^{-2}$ is also achieved.

## Results

### Construction of 2D VMT nanofluidic channel

VMT is a naturally occurring 2:1-type aluminosilicate mineral with a layered crystal structure, comprised of one aluminum-oxygen or magnesium-oxygen octahedral sheet sandwiched by two opposing silicon-oxygen tetrahedral sheets (Supplementary Figs. 1 and 2)[13,14]. Here, the monolayer VMT nanosheets were obtained using an eco-friendly ion-exchange preparation strategy (Fig. 1a). Initially, $Na^+$ ions were exchanged with pre-existing hydrated cations located within the interlayer region of VMT, followed by a complete exchange using the LiCl solution. This process resulted in intercalation of $Li^+$ ions within the interlayer region, causing an expansion of the crystal structure, ultimately leading to an increased interlayer spacing. After the two-step ion-exchange process, the interaction force between adjacent VMT layers was greatly reduced. Subsequently, $H_2O_2$ was introduced between the VMT layers, generating a large number of bubbles from the thermal decomposition of $H_2O_2$, which directly delaminated the VMT particle through mechanical force. The resulting VMT nanosheets were well dispersed in water and exhibited the typical Tyndall effect (Supplementary Fig. 3). Atomic force microscopy (AFM) and transmission electron microscopy (TEM) images (Fig. 1b and Supplementary Fig. 4) indicated the VMT nanosheets had a smooth surface, and an intact structure with a thickness of approximately 1.5 nm. As shown in Supplementary Fig. 5, the scanning electron microscopy (SEM) images confirmed large lateral dimensions of VMT nanosheets and optical microscopy and Image J software further determined the lateral size of VMT nanosheets to be approximately 12 μm (Fig.1c).

The VMT nanochannel membrane was prepared by parallelly stacking the as-obtained VMT nanosheets on PVDF porous substrate using a vacuum-assisted filtration method (Supplementary Figs. 6 and 7). The porous surface of PVDF was entirely covered by the deposition of VMT nanosheets, with no apparent voids or defects on the surface of the VMT layer. SEM images (Fig. 1d) and the X-ray diffraction (XRD) patterns (Supplementary Fig. 8) revealed the layered structure of the VMT membrane cross-section and a uniform membrane surface, energy dispersive spectroscopy (EDS) analysis also demonstrated a homogeneous stacking of VMT nanosheets during membrane fabrication (Supplementary Figs. 9 and 10). The VMT membrane exhibited great mechanical properties and flexibility with a tensile strength of approximately 38 MPa and a fracture strain of 2.52% (Fig. 1e). Flexibility is also a pivotal mechanical attribute for practical applications in osmotic energy conversion. As depicted in our images (Supplementary Fig. 11), the VMT membrane exhibits excellent flexibility and remains structurally intact even after undergoing multiple folds.

In our study, the VMT membranes with nano-scale thickness were attempt to prepared to achieve a high ion flux. Here, the prepared VMT membrane was transferred to a silicon wafer with a flat surface and the thickness of the membrane was precisely quantified by AFM (Fig. 1f) and the ultrathin VMT membrane show a uniform thickness distribution. In the case of a thin VMT membrane, the appearance of wrinkles on the membrane surface may be attributed to the unevenness of the porous substrate. However, the structure of the VMT membrane remained intact and no holes or other defects were visible. As the quantity of deposited VMT nanosheets increases, they effectively coat the substrate surface, resulting in a flat morphology due to the uniform stacking of nanosheets (Supplementary Fig. 12). Fourier-transform infrared (FTIR) spectra were utilized to identify the functional groups of the VMT nanochannel membrane. The spectra, as depicted in Supplementary Fig. 13, revealed peaks observed at 454 $cm^{-1}$ and 1000 $cm^{-1}$, which could be attributed to the asymmetric stretching vibrations of Si−O. The presence of the absorbance peak at approximately 688 $cm^{-1}$ was assigned to M−O−Si (M = Fe, Al, Mg) plane deformation vibrations, and the broad absorption peaks of near 1640 $cm^{-1}$ and 3400 $cm^{-1}$ were ascribed to −OH vibrations. X-ray photoelectron spectroscopy (XPS)

indicated the presence of constitutive elements Mg, Al, and Si characteristic peaks with bonding energy at 1303 eV, 74 eV, and 103.5 eV. The O 1$s$ peaks at 532.8 eV and 530.8 eV represented the Si-O bonds and the surface hydroxyl groups, respectively.

In our pursuit of membrane scale-up preparation, we sought to leverage vacuum-assisted filtration, a widely adopted method in the realm of 2D membrane preparation[23,24], to increase the membrane size by expanding the filtration area from 12.56 cm² to 78.5 cm².

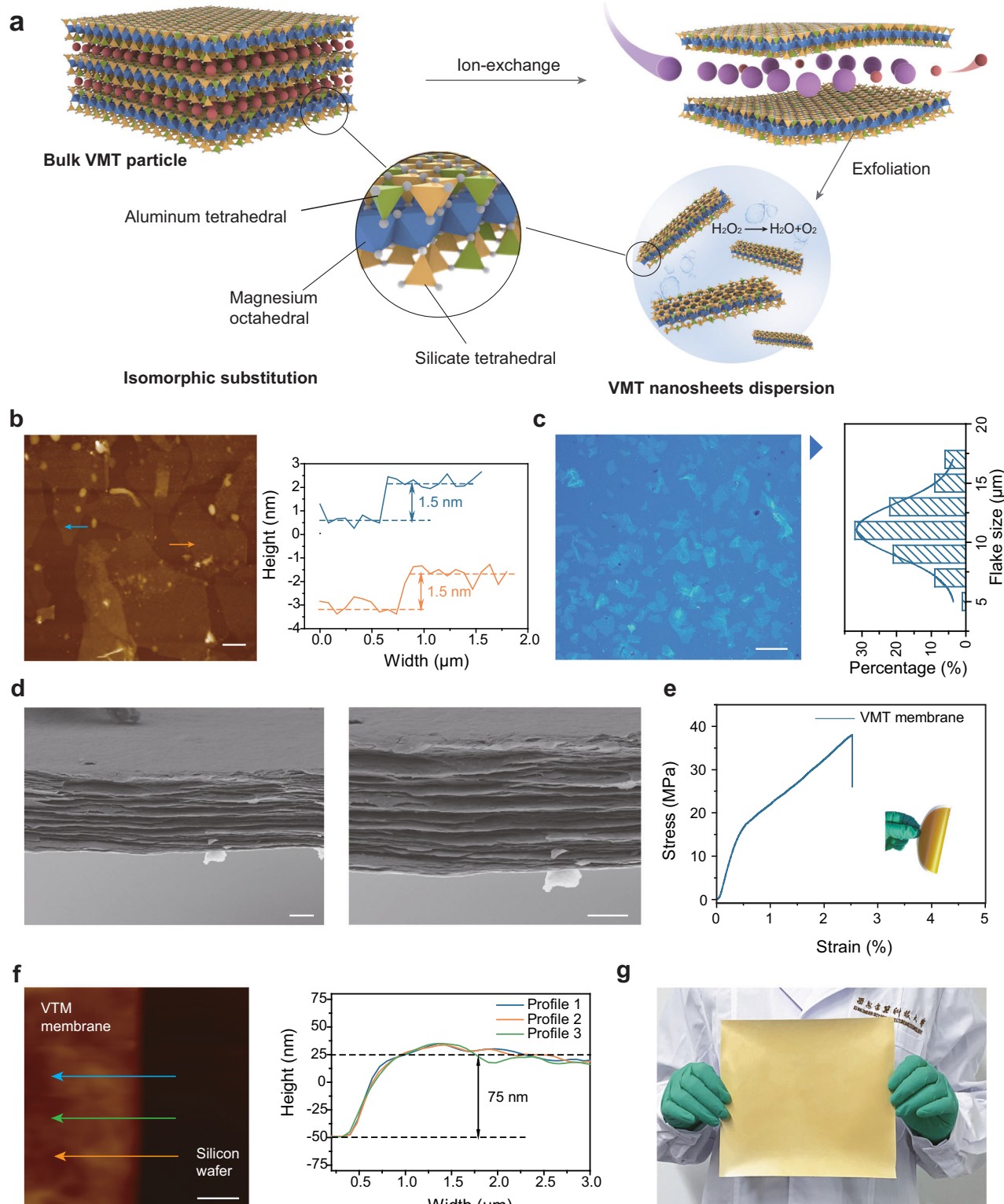

**Fig. 1 | Fabrication and characterization of the VMT nanosheets and VMT membrane. a** Schematic illustration of the preparation of the large-scale VMT nanosheets. **b** AFM image and height profile of the nanosheets corresponding to the AFM image. Scale bar, 2 μm. **c** Optical microscopy image of the large-scale VMT nanosheets. Scale bar, 20 μm. **d** SEM image of the cross-sectional morphology of the VMT membrane. Scale bar, 3 μm. **e** Stress–strain curve of VMT membrane. **f** AFM image of the VMT membrane and height profile of the VMT membrane corresponding to the AFM image. Scale bar, 1 μm. **g** Optical image of a VMT membrane.

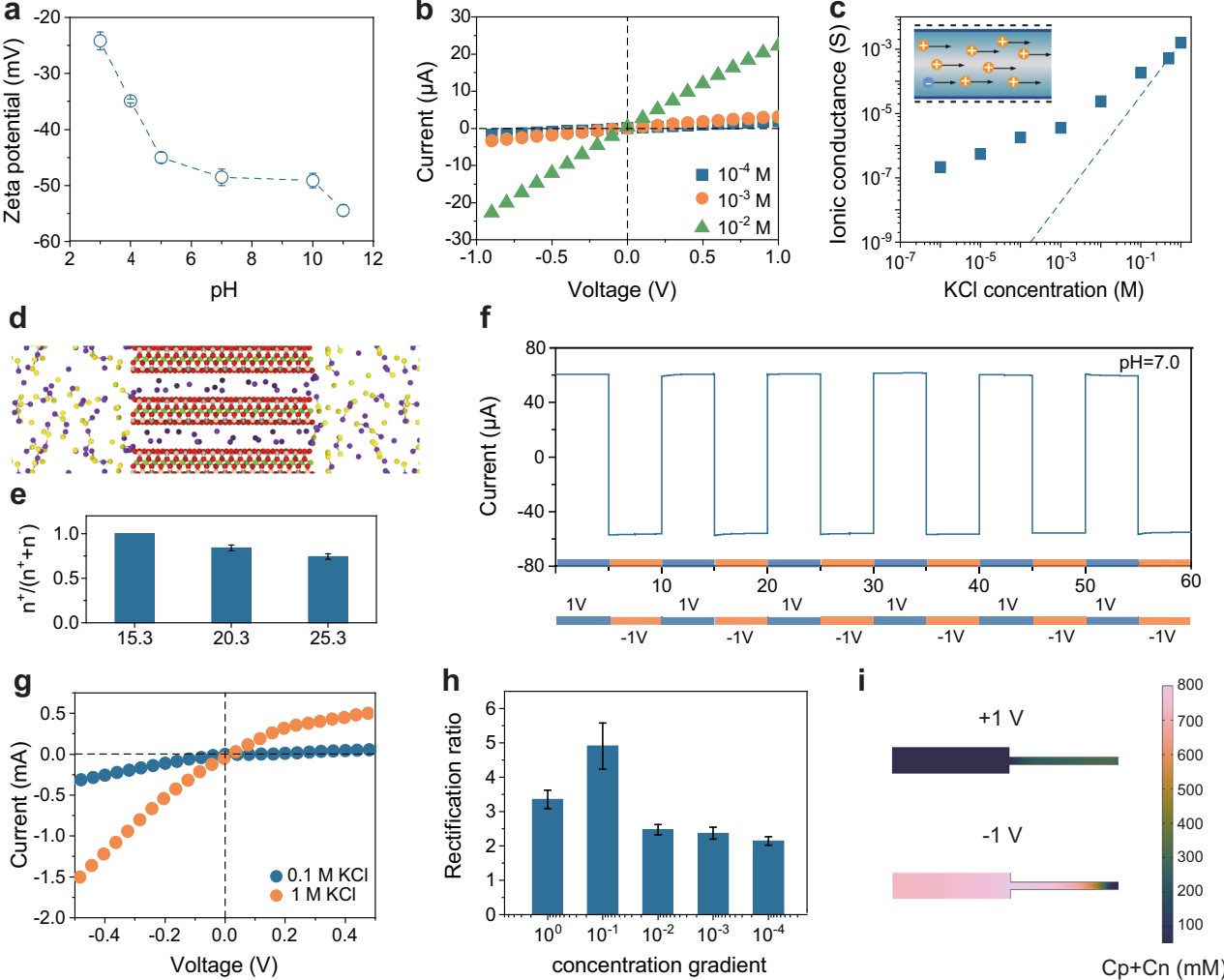

**Fig. 2 | The ionic transmembrane properties of the VMT-based membrane.**
**a** Zeta potential of VMT nanosheet colloidal suspension at different pH. **b** Representative I–V curves of the VMT membrane. **c** Conductivity of VMT membrane as a function of salt concentration. **d** Classical molecular dynamics simulation snapshot that showed the distribution of ions. **e** The ratio of cation (η) in nanochannel. **f** I–T curves of VMT membrane in 0.1 M KCl at pH 7 with an external bias alternating between +1 and −1 V. **g** I–V curves of the VMT-PVDF membrane measured in electrolyte solutions with different concentrations. The applied transmembrane voltage was between −0.5 V and +0.5 V. **h** Ionic rectification ratio under different concentrations calculated from the I–V curves. **i** Theoretical calculation of ionic concentration distribution in nanochannel at −1 V and +1 V. Error bars indicated the standard deviations from three different samples.

Furthermore, given the low efficiency and time consuming of vacuum-assisted filtration, we explored a more efficient spray coating method to prepare a large-scale VMT membrane, providing a clearer illustration of scalability. As evidenced in Supplementary Figs. 14 and 15, the 2D VMT membrane, covering an area of 300 cm² and with a thickness of 3 μm, was successfully prepared within 30 min. The resulting membrane displayed a uniformly consistent surface, devoid of any discernible structural defects (Fig. 1g). By employing this approach, the VMT membrane exhibits the potential for facile expansion, making it a promising candidate for large-scale manufacturing.

**Selective ion transport behavior in VMT-based membrane**
In our study, it was shown that the zeta potential of VMT nanosheets was approximately −50 mV at pH 7 (Fig. 2a), which was a relatively low value compared to other 2D materials. For instance, the zeta potential of graphene oxide (GO)[25] nanosheets was approximately −42 mV due to the abundant carboxyl group, while the zeta potential of Ti₃C₂Tx[26] nanosheets was around −40 mV as a result of the deprotonation of surface hydroxyl groups. The surface charge density ($\sigma_s$) of VMT

nanosheets was also calculated from the zeta potential value using the Gouy–Chapman equation as follows[27]:

$$\sigma_s = \frac{2\varepsilon kT\kappa}{ze} \sinh\left(\frac{Ze\zeta}{2kT}\right) \quad (1)$$

where $\varepsilon$, $z$ and $e$ refer to the dielectric constant, valence of the counterions, and elementary charge, respectively and $\kappa$ represents the reciprocal of the Debye length, which is given by the expression $\sqrt{I}/0.304$. Where $I$ is the ionic strength defined as $(1/2)\sum z_i^2[x_i]$, and $x_i$ is the molar concentration of the $i$ species, and $z_i$ is ionic valence. The VMT nanosheets, when dispersed in a 0.01 M KCl solution at pH = 7, exhibited a stabilized zeta potential of −50 mV, with a calculated surface charge density of −13.4 mC m⁻².

Such a high negative charged surface property can be attributed to the unique structure of VMT, which could be discussed in the following two aspects. Firstly, an increasing number of hydroxyl groups emerge on the surface and edge of VMT due to the broken bonds at the siloxane group during the exfoliation process into monolayer VMT nanosheets as confirmed by FTIR (Supplementary Fig. 13)[28,29]. When

immersed into an electrolyte solution, the deprotonation of these terminal groups results in the negatively charged surface of the VMT nanosheet. Such a deprotonation process is reinforced as the pH increases, as confirmed by the measured zeta potential at various pH levels. Furthermore, the isomorphic substitutions of $Al^{3+}$ for $Si^{4+}$ in tetrahedral layers and $Mg^{2+}$ or $Fe^{2+}$ for $Al^{3+}$ in octahedral layers also contribute to the negative surface charges[30]. Thus, despite the fact that an acidic environment suppresses the deprotonation of the surface hydroxyl group, the zeta potential of VMT nanosheets maintains its negatively charged surface even when the pH decreases to 2, due to these permanent charges.

To investigate the selective ionic transport properties of the VMT nanochannel, the ionic current−voltage (I–V) curves were first investigated with a freestanding VMT lamellar membrane using KCl electrolyte at different concentrations (Fig. 2b). The membrane was positioned in the center of a two-compartment conductance cell using two poly-tetra-fluoroethylene (PTFE) gaskets, and transmembrane current was measured using a pair of homemade Ag/AgCl electrodes.

Due to the significant negative charges presented on the surface of VMT, an EDL was formed at the interface of the charged VMT nanosheets in the KCl electrolyte solution. In this region, the local electro-neutrality is not obeyed and the ion concentration profile departs from their bulk values due to the electrostatic interaction of cations and anion with the surface charge. Based on the XRD results, the interlayer spacing of VMT membrane was approximately 1.53 nm. The thickness of VMT ($h$) is defined by geometric thickness (Supplementary Fig. 16), $h = d_{T-B} + r_T + r_B$, where $d_{T-B}$ is the sum of the vertical distance between the topmost and bottommost atoms, $r_T$ and $r_B$ is the van der Waals radius of the topmost and bottommost atoms, respectively[31]. The $h$ can be obtained by performing the DFT calculation of structure relaxation that $h = 9.6$ Å. Based on this, the height of the nanochannel can be estimated to be 0.6 nm. According to the EDL theory, the Debye length represents the approximate distance over which the potential at a charged surface dissipates into the bulk solution. This length is closely linked to electrolyte concentration and valence and can be theoretically calculated using the equation[32]:

$$\lambda_D = \sqrt{\frac{\varepsilon RT}{\sum_{i=1}^{N} F^2 Z_i^2 C_{i,0}}} \qquad (2)$$

Here, $\varepsilon$ is the solution permittivity, $R$, and $T$ are the gas constant and temperature, respectively; $F$ is the Faraday constant; $N$ is the total number of ionic species; $Z_i$ and $C_{i,O}$ are the valence and bulk concentration of $i$th ionic species, respectively. It could be observed that with increasing ion concentration, the Debye length decreases. For example, in a 0.1 M KCl solution, the Debye length is approximately 1 nm, whereas in a 1 M KCl solution, it reduces to about 0.3 nm. Based on calculation of VMT channel diameter, it could be inferred that when the KCl concentration falls below 0.1 M, the EDLs on opposite surfaces overlap within the nanochannels. Consequently, the ion distribution and population become significantly influenced by the surface charges, owing to the robust electrostatic interaction. As a result, at high concentrations, the corresponding transmembrane ionic conductance demonstrated a linear correlation with solution concentration, similar to bulk behavior. However, a deviation from bulk ionic conductance was observed when the electrolyte concentration dropped below 0.1 M as shown in Fig. 2c. Such a phenomenon of ionic transport was consistent with the typical surface-charge-governed ion diffusion behavior observed in previous reports, which was attributed to the fact that ion diffusion was significantly impacted by surface charges when the channel size was reduced to nano-scale[33,34]. It should be noted that the Debye length provides a qualitative depiction of the spatial extent of the electrostatic effects within the EDL. However, in highly confined channels, factors such as ion size and interactions between ions

become crucial. In conventional Poisson−Boltzmann theory, ions within the EDL are commonly treated as point charges, neglecting these intricate details[35].

The classical MD simulation (see details in "Methods" section and Supplementary Data 1) was also performed to further analyze the surface-charge-governed ion transport in VMT lamellar membrane under the high-concentration environment (Fig. 2d and Supplementary Fig. 17). It could be observed that $K^+$ was still the dominant charge carrier in the VMT nanochannel, while $Cl^-$ was almost excluded due to strong electrostatic repulsion. However, the ratio of cation ($\eta$) in nanochannel decreased from 1.0 to $0.74 \pm 0.03$ when the height of channel ($h$) increased to 2.53 nm, and ion pairs, which were associations of cations and anions in high-concentration solutions, could be found in the expanded VMT channels (Fig. 2e).

These results highlight that benefiting from the unique surface charge property and nano-scale diameter of VMT channel, an efficiently distinguishing ions with different charge polarities was achieved, even under the high saline environment. The ion distribution on the VMT membrane after immersing into the KCl solution for 10 h obtained by EDS tests also confirmed the preferential selectivity of negatively charged VMT surface for $K^+$ due to strong electrostatic effect. The content information corresponding to the mapping images in Supplementary Fig. 18 indicates a significantly higher amount of $K^+$ compared to $Cl^-$, further confirming the VMT membrane's pronounced cation selectivity. To evaluate the stability of ion transport through the VMT membrane, a current−time (I–T) test was conducted by alternately applying an external bias voltage of +1 V/−1 V. Each test cycle lasted for 10 min and was repeated 6 times without interruption in 0.1 M KCl solution. As illustrated in Fig. 2f, the currents remained essentially constant under both positive and negative bias, respectively, indicating the excellent ion transport stability of the VMT membrane.

Adequate chemical resistance and mechanical stability in harsh environments are crucial properties for membranes in practical osmotic energy harvesting applications. The VMT membrane showed adequate chemical resistance and robust mechanical stability when subjected to challenging environmental conditions, as demonstrated by its structural integrity after immersion in base or acid for 7 days (Supplementary Fig. 19). Chemical resistance at high temperatures is crucial for practical osmotic energy applications, especially concerning industrial wastewater. Previous study demonstrated that VMT nanochannels can retain their layered structure and maintain their proton conduction functions even after annealing at 500 °C in air, showcasing their extraordinary thermal stability[14]. We also conducted experiments to analyze the ion transport behavior in VMT-based nanofluidic membranes after immersion in acidic and alkaline solutions with a concentration of 0.001 M at 80 °C for 6 h, respectively. We confirmed that the employed immersion treatment had almost no impact on the surface-charge-governed ion transport behavior in the VMT nanochannels membrane, indicating the chemical resistance of the VMT membrane at high temperatures (Supplementary Fig. 20).

The remarkable ion selectivity and ion transport stability of the 2D VMT membrane provides a solid basis for efficient osmotic energy conversion. In our previous investigation, the laminar VMT membrane exhibited effective rejection of dye molecules, even when its thickness was reduced to 15 nm[36]. This behavior was mainly primarily because the lateral size of VMT nanosheet could eliminate unsatisfactory occasional defects during nanosheet stacking for the membrane with nano-scale thickness. To further enhance the practical application potential of the VMT membrane in osmotic energy harvesting, we designed a composite VMT-PVDF membrane by combining an ultrathin VMT membrane as the separation layer with a PVDF membrane that had an average pore size of 0.1 μm as the porous substrate. In this composite configuration, the PVDF substrate offers adequate mechanical strength for long-term utilization, while the ultrathin VMT

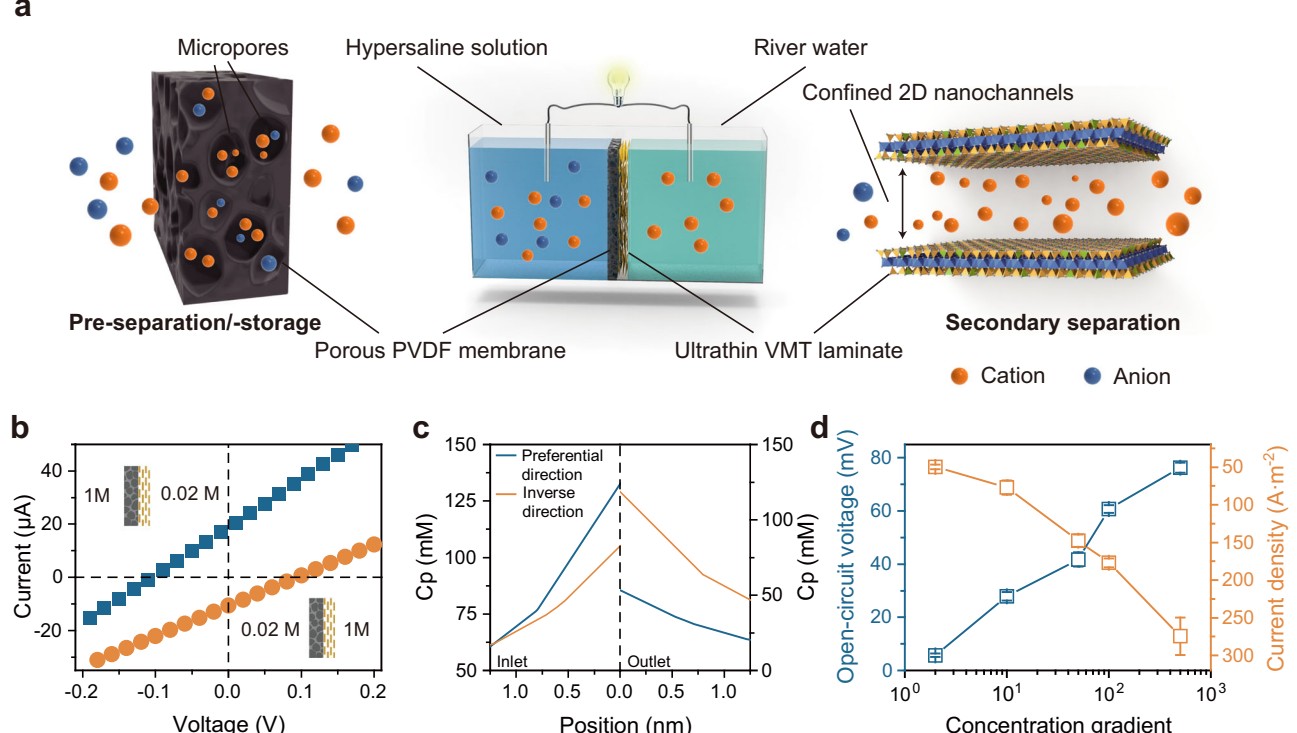

**Fig. 3 | Selective ion transport behavior in VMT-based membrane. a** Schematic illustration of dual-separation mechanism. **b** Two configurations under a 50-fold concentration gradient with $C_{high}$ = 1 M KCl, $C_{low}$ = 0.02 M KCl. **c** Numerical simulations depicting the cation concentration profiles on both sides of the VMT layer, which were plotted along the dashed line near the orifice of the VMT channel (Supplementary Fig. 21). **d** The open-circuit potential and short-circuit current of the VMT-PVDF membrane with increasing concentration gradient. The high-concentration side of the electrolyte solution was fixed as 1 M on the PVDF side and the low-concentration solution on the VMT side varied from 0.002 M to 0.5 M. Error bars indicated the standard deviations from three different samples.

laminates with minimized diffusion pathways ensure high ion flux with exceptional ion selectivity. As previously explained, the thickness of the VMT lamellar membrane can be accurately measured using AFM. In contrast to the homogeneous VMT membrane that displayed a linear current−voltage response, ion rectification behavior was observed in the VMT-PVDF composite membrane with an asymmetric structure property (Fig. 2g, h). The ionic current rectification (ICR) ratio, defined as the ratio of the current values under ±1 V, reached the highest ICR ratio value of 5.44 for the VMT-PVDF membrane. Based on these findings, it can be assumed that the PVDF membrane not only provided mechanical support but also played a role in constructing an ionic diode where ions preferentially migrated in one direction, while backward transport was restricted.

To gain a deeper understanding of the rectification effect in the VMT-PVDF system, a numerical simulation based on the Poisson and Nernst−Planck (PNP) model was conducted (Supplementary Fig. 21)[37]. As shown in Fig. 2i and Supplementary Fig. 22, significant accumulation or depletion of ions occurred at the junction region under different bias polarities (±1 V). Specifically, when a negative bias was applied, the amount of cation flux into the nanochannel was larger than that flowing out of the nanochannel, leading to a relatively high ionic current as steady-state was reached. Conversely, when a positive bias was applied, more ions left the channel, causing the ionic conductance to decrease. It is noteworthy that the ionic rectifying effect can be observed in many nanofluidic devices with the asymmetric channel geometric structure. However, prior studies have shown that this rectification effect is not apparent in the high-concentration range due to the attenuated EDL range[38–40]. In comparison, the VMT-based nanofluidic system showed maximum ICR ratio in higher concentration ranges and reached the peak value at 0.1 M KCl.

To investigate the efficacy of the VMT-PVDF membrane in osmotic energy conversion, we conducted I−V measurements in the presence of a salinity gradient. When the concentration of the VMT solution was 0.02 M and the concentration of the PVDF solution was 1 M ($C_{VMT}$/$C_{PVDF}$ = 0.02 M/1 M), the short-circuit current ($I_{SC}$) and the open-circuit potential ($V_{OC}$) corresponding to zero external bias and zero current, respectively, were found to be 110 mV and 18.8 μA, respectively. However, we observed different ion transport behaviors when the high concentration was oppositely set on the VMT side with the same salinity gradient ($C_{VMT}$/$C_{PVDF}$ = 1 M/0.02 M). Here, a decrease in both $V_{OC}$ and $I_{SC}$ to 90 mV and 10.7 μA, respectively, and an increase in the inner resistance of the VMT-PVDF membrane by approximately 45% could be observed. To elucidate the different ion transport behaviors under the two concentration gradient configurations, we divided the ion transport process through the VMT-PVDF membrane into two stages (Fig. 3a, b). When the high concentration of the KCl solution was on the PVDF side, the negatively charged porous PVDF membrane served as an ion-storage space, capturing a large number of cations into the inner macro-pores, thereby achieving the pre-separation of cations and ion storage. Due to this ion-storage mechanism, when the ions attempted to diffuse through the VMT selective layer to complete the transmembrane transport, the higher chemical potential gradient led to a higher ion flux because the actual transmembrane transport concentration gradient was higher than the original one.

To verify the above speculation, a numerical simulation study was also conducted. Figure 3c depicted the cation concentration profiles on both sides of the VMT layer, which was plotted along the dashed line near the orifice of the VMT channel. In the case that the high-concentration solution was set on the PVDF side, the concentration of cations at the high-concentration end (inlet side) was significantly

higher than the bulk solution, and maintained a lower cation concentration at the outlet side compared to the case in the inverse direction, increasing the real salinity gradient by threefold and ultimately promoting the diffusion current in this direction. In contrast, when $C_{VMT}/C_{PVDF} = 1 M/0.02 M$, a significant increase in cation concentration at the low-concentration end (outlet side) impaired the effective concentration difference, and ion transport under this gradient configuration required overcoming a relatively higher energy barrier. Based on these findings, the high-concentration solution is set on the PVDF side in the subsequent studies on osmotic energy conversion performance.

The I−V curves of VMT-PVDF were investigated under various salinity gradients. The concentration of KCl solution on the PVDF side remained fixed at 1 M during the measurement, while the low-concentration solution on the VMT side varied between 0.002 M and 0.5 M KCl. It is noteworthy that the $V_{OC}$ is comprised of two components, namely, the diffusion potential ($E_{diff}$) derived from the membrane and the redox potential ($E_{redox}$) generated by the unequal potential drop of the electrode-solution interface at different electrolyte concentrations, as demonstrated in the equivalent circuit. To obtain the actual $E_{diff}$ and $I_{diff}$, a pair of agarose salt bridges were employed to eliminate the redox potential (Supplementary Fig. 23). The results presented in Fig. 3d revealed that both current density and diffusion potential values gradually increased with increasing salinity gradient, reaching approximately 285 A/m² and 78.2 mV, respectively. The ion selectivity of the membrane is commonly quantitatively characterized by the cation transfer number ($t_+$), which indicates perfect cation selectivity when equal to 1. The VMT-PVDF membrane demonstrated remarkable cation selectivity as evidenced by its calculated $t_+$. Prior research has shown that the ion selectivity of membranes tends to decline in concentrated solutions owing to the diminishing range of electrostatic interactions and screening effects[41,42]. Interestingly, the concentration of the electrolyte solution did not have a significant impact on the ion selectivity of the composite membrane. Remarkably, even when the concentration of the electrolyte solution on the high side reached 1 M, the value of $t_+$ remained steady at 0.8 (Supplementary Fig. 24).

Based on the above results obtained from MD analysis, it can be concluded that the VMT nanochannel exhibits exceptional ion selectivity even when subjected to high-salinity gradient conditions. Furthermore, in the VMT-PVDF composite system, the VMT component plays a more significant role in determining the overall selectivity. According to the numerical calculations based on the PNP model, when the charge density of the PVDF component was held constant, and the charge density of the VMT layer was varied, it was observed that an increase in the surface charge density of the VMT section results in an apparent increase in the $t_+$. Moreover, the incorporation of the PVDF component in the composite system also showed a positive impact on improving the total ion selectivity, and the heterogeneous system demonstrated $t_+$ value was 7.2% higher than that of a single VMT channel at a salinity gradient of 500 (Supplementary Figs. 25 and 26).

## Osmotic power conversion performance of VMT-based membrane

To assess the potential of the VMT-based nanofluidic device for osmotic energy conversion, the output power density was measured to determine if it could be harvested and transferred to an external circuit to supply an electrical load resistor. In our experiments, the high-salinity (NaCl) concentration was set to 0.5 M on the PVDF side, while the low-salinity side concentration was set to 0.01 M. The output electric power (P) was calculated using the equation $P = I^2 \times R_L$, where I represented the measured current and $R_L$ referred to load resistance. It was observed that the current density decreased with increasing load resistance, and the maximum output power density ($P_{max}$) was

achieved when the external load resistance was nearly equal to the internal resistance of the membrane (Fig. 4a, b). To further investigate the effect of membrane thickness on osmotic energy generation, the output power density was measured at various thicknesses. As illustrated in Supplementary Fig. 27, a decrease in VMT membrane thickness from 100 nm to 10 nm led to an increase in ion flux due to the decrease in resistance, resulting in better energy conversion performance. However, further reduction in thickness resulted in structural defects that led to low selectivity, thereby resulting in a lower output power density. It was observed that the maximum output power density of 5.45 W m⁻² was achieved when the ion flux and selectivity effect were balanced at a VMT membrane thickness of 30 nm. This power density exceeded the commercial benchmark of 5.0 W m⁻² under seawater/freshwater systems, indicating that the RED system based on the ultrathin VMT membrane has met the industrial requirements of osmotic energy harvesting.

To evaluate the capability of the VMT-PVDF membrane to maintain superior osmotic energy conversion performance under high-salinity gradient conditions, we held the low-concentration solution constant at 0.01 M NaCl on the VMT side and varied the concentration of the high-salinity solution from 0.05 to 5 M NaCl. With increasing salinity gradient, the transmembrane driving force was amplified, leading to more efficient directional diffusion of cations and a rise in the output power density. A maximum power density of 33.76 W m⁻² was attained at a salinity gradient of 500-fold and sustained at around 30 W m⁻² for 216 h without significant degradation (Fig. 4c), highlighting the maintenance of such high energy harvesting performance under long-term working conditions. In previous studies, we observed a non-linear correlation between power density and salinity enhancement multiplier, as well as a high reduced rate of power density enhancement at higher concentration levels. Our investigation revealed an impressive 6.2-fold increase in $P_{max}$ in response to a 10-fold escalation in salinity gradient. These findings signify that the energy conversion performance of the VMT-PVDF confined channel system is not compromised under hypersaline conditions, surpassing the performance of existing nanofluidic membranes under such challenging conditions as shown in Fig. 4d.

In addition to salinity gradient, ion type and valence were also important factors to be considered in practical osmosis energy harvest applications. We chose several typical cations to further investigate the osmotic energy conversion performance of VMT-PVDF membrane in different electrolyte environments. As shown in Fig. 4e and Supplementary Fig. 28, under a 50-fold salinity gradient ($C_{PVDF}/C_{VMT} = 0.5 M/0.01 M$), the values of output power density follow the trend of $KCl > NaCl > LiCl > MgCl_2$ and the highest value of 7.13 W m⁻² was obtained in the KCl electrolyte solution. The ab-initio molecular dynamics (AIMD) calculation for the diffusion of $Li^+$, $K^+$, and $Mg^{2+}$ in VMT nanochannel was performed to explain this order first (Supplementary Fig. 29 and Supplementary Data 2). We found that $K^+$ and $Li^+$ exhibited a strong preference for adsorption on the surface of the basal plane while $Mg^{2+}$ located in the middle of the channel (Fig. 4f), which could be explained by the hydration strength that monovalent cation is much smaller than divalent cations (Supplementary Table 3) and thus attached with more flexible hydration shells (HSs). The mean-square distance (MSD) curves of ions (Fig. 4g) showed the order of diffusion follows $K^+ > Li^+ > Mg^{2+}$. The difference in the diffusion for monovalent cations could be inferred from the different diffusion paths that $K^+$ migrated among the center of siloxane rings mainly while $Li^+$ migrated near the oxygen groups of siloxane rings (Supplementary Figs. 30 and 31, and see details in Supplementary Movies). The near-surface diffusion of monovalent cations made faster diffusion than $Mg^{2+}$, the latter migrated with tight HSs in the narrow nanochannel. On the other hand, dehydration when hydrated ions entered the nanoscale channels made different energy barriers that multivalent cations maintained tight HSs and thus needed to overcome higher barriers

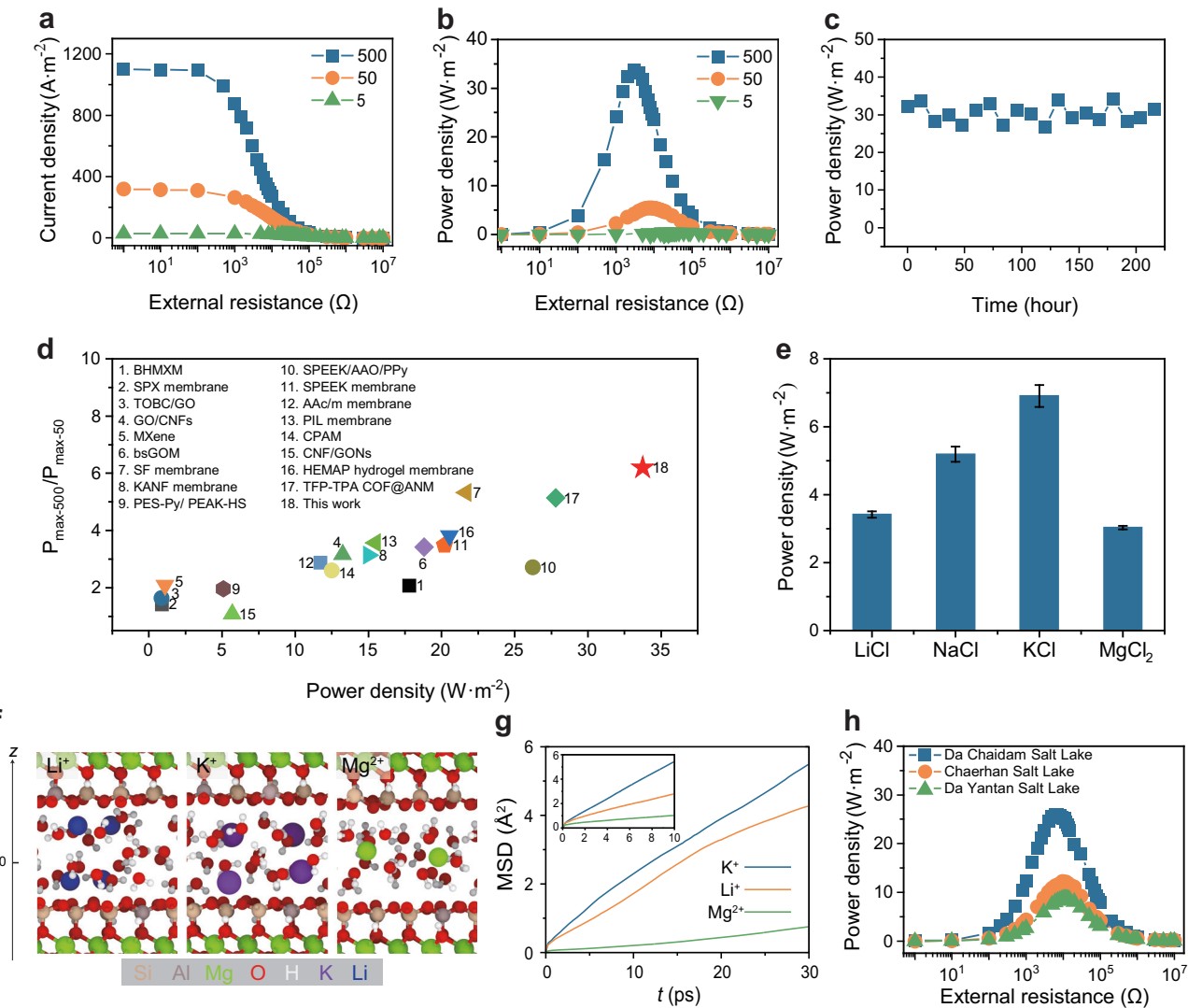

**Fig. 4 | High-performance osmotic energy conversion of the VMT-based membrane. a**, **b** Current density and output power density of VMT-based membrane as functions of load resistance under three salinity gradients. The low-salinity solution was placed in the VMT side and fixed at 0.01 M NaCl. High-salinity solution was tunable from 0.05 to 5 M NaCl. **c** The long-term stability of the energy conversion at 500-fold salinity gradient. **d** Comparison with previous studies (details in Supplementary Table 2). The ratio of $P_{max}$ at 500-fold and 50-fold salinity gradient ($P_{max-500}/P_{max-50}$) and power densities of the membranes. **e** Output power density under different types of electrolytes. **f** The AIMD snapshots that showed the distribution of ions in VMT nanochannel. The interlayer spacing was 1.53 nm, which was the same as the XRD results (Supplementary Fig. 8). **g** The MSD curves of ions. The inset showed the MSD curves of water. The value of diffusion coefficient was proportion to the slope of MSD. **h** Power density under three different highly concentrated natural brines. Error bars indicated the standard deviations from three different samples.

than monovalent cations, which led to slow migration (Supplementary Fig. 32). It should be noted that considering the different valences of cations, the ionic current generated by the transport of a specific quantity of divalent $Mg^{2+}$ through the nanochannels equals the ionic current produced by twice the amount of $Li^+$ passing through these channels. As a result, our AIMD simulation revealed a higher transmembrane permeation rate for $Li^+$ compared to $Mg^{2+}$, however, the power density in the $MgCl_2$ solution system is comparable to that in the LiCl solution system. This observation aligns with previous studies investigating the impact of cation type on output power density[18,43].

Based on the aforementioned discussions, the VMT-PVDF membrane displays remarkable potential for osmotic energy conversion, particularly in high-salinity environments, while maintaining excellent ion selectivity. To assess its potential in real-world hypersaline applications, practical salt-lake−freshwater osmotic energy harvesting systems were established using three different highly concentrated

natural brines, including Da Chaidam Salt Lake, Chaerhan Salt Lake, and Da Yantan Salt Lake. These systems achieved high output power densities, with Da Chaidam Salt Lake exhibiting a value of 25.9 W m$^{-2}$ (Fig. 4h), far surpassing those reported in the literature. Notably, although the total salinity of the three brines was identical (total salt concentration of 5 M), they exhibited different power densities. Comparison of the ionic composition of the three brines (Supplementary Table 4) revealed that this was due to the significantly higher NaCl content in the water of Da Chaidam Salt Lake than in the Cold Lake brine, while the $MgCl_2$ content was relatively low. This finding aligns with our previous research, which demonstrated that the output power density was closely related to the cation species. This discovery aligns with our above discussion, which has demonstrated the close relationship between the output power density and cation species, providing insights into the potential of VMT-PVDF membranes in practical hypersaline environments.

## Discussion

In summary, we have established a heterogeneous nanofluidic system based on the large-scale VMT nanosheets via simple scalable method, realizing a dual-separation mechanism where cations are initially concentrated in the negatively charged porous substrate, followed by further separation with high ion flux in the ultrathin VMT laminate. The heterogeneous nanochannel membrane with environmentally friendly preparation method showed great potential for industrialization and exceptional osmotic energy harvesting performance, even in hypersaline systems. A remarkable osmotic conversion power density of 33.76 W m$^{-2}$ at a 500-fold salinity gradient was achieved, with a 6.2-fold increase in P$_{max}$ for a 10-fold increase in salinity gradient, making the VMT-PVDF membrane stand out from the state-of-the-art nanochannel membranes. Ab-initio molecular dynamics simulations confirmed the selective ion transport mechanism in VMT-based nanofluidic system. Furthermore, we tested our VMT-PVDF heterogeneous membrane using natural salt-lake brines, with the highest power output density obtained from Da Chaidam Salt Lake at 25.9 W m$^{-2}$. This work provides valuable guidance for designing advanced nanochannel membranes to capture osmotic energy from practical high-salinity resources, such as industrial wastewater, natural seawater, and brine.

## Methods

### Materials and reagents

The thermally expanded vermiculite (VMT) crystals were obtained from Henan province, China. Hydrogen peroxide (H$_2$O$_2$, 30%), lithium chloride (LiCl AR, ≥99%), sodium chloride (NaCl, AR, ≥99.5%), potassium chloride (KCl, AR, ≥99.5%), and magnesium chloride (MgCl$_2$, AR, ≥98%) were purchased from Tianjin Kemiou Chemical Reagent Co., Ltd, China and were used as received without further purification.

### Synthesis of large-scale VMT nanosheets

Monolayer VMT nanosheets with large lateral size were synthesized via a two-step ion-exchange exfoliation process[14]. Initially, 3 g of thermally expanded VMT powder was immersed in 200 mL of saturated NaCl solution and refluxed at 120 °C with continuous stirring for 24 h. The obtained precipitate, designated as Na-VMT, was washed with deionized (DI) water. Next, Na-VMT was dispersed in 200 mL of a 2 M LiCl solution, followed by reflux at 120 °C for 24 h to ensure complete ion exchange. Thorough rinsing with DI water was conducted to eliminate residual LiCl, yielding the desired Li-VMT product. Subsequently, the Li-VMT product was immersed in 100 ml of a 30% H$_2$O$_2$ solution, which further delaminated it into the VMT nanosheets. High-speed centrifugation (7100 g, 30 min) was carried out to effectively remove impurities and multilayered nanosheets, and the resulting sediment was redispersed. Low-speed centrifugation (690 g, 30 min) was then performed to collect the stably dispersed large-scale VMT flakes in the upper solution The final product was a stable light-yellow suspension of monolayer VMT nanosheets with large dimensions, which were further characterized and analyzed. Membrane with a thickness of 1 μm was prepared by depositing 0.8 mg cm$^{-2}$ of nanosheets on the substrate and subsequently the obtained membranes were dried overnight under ambient conditions.

### Preparation of the VMT-based membranes

The VMT-PVDF heterogeneous membrane was fabricated using a vacuum filtration process, wherein the nanosheets were deposited onto a substrate using a dilute colloidal solution (0.001 mg ml$^{-1}$). To prepare the freestanding VMT lamellar membrane for investigating intrinsic ion selectivity, a mixed cellulose membrane with an average pore size of 0.22 μm was chosen as the substrate, as the VMT membrane could be carefully peeled off from the support membrane. To construct the VMT-based heterogeneous nanofluidic system, a porous PVDF substrate with an average pore size of 0.1 μm was selected, and

the VMT laminates were anchored onto the PVDF substrate firmly to evaluate its potential for osmotic energy harvesting in a hypersaline environment.

For the scalability of VMT membrane area, VMT nanosheets dispersion was formed on the fixed Polyethersulfone (PES) substrate using a homemade spray coating device. The distance between the spray gun and the substrate was set at 20 cm while nitrogen was used as a carrier gas. The spray gun was moved at a speed of 45 mm s$^{-1}$ to obtain uniform VMT membrane.

### Characterization

The VMT crystals and the d-spacing of the VMT membrane were characterized by X-ray diffraction (XRD, UltimqlV, Japan) with Cu Kα radiation at a step of 0.02° and a collection time of 5°/step. The microstructures and elemental distribution characteristics of the nanosheets and membranes were measured by scanning electron microscope (SEM Zeiss Gemini 300, Germany). The morphology of the VMT nanosheets was characterized by transmission electron microscopy (TEM Talos F200X, USA). Atomic force microscopy (AFM Bruker Multimode 8) was used to obtain morphological images of the nanosheets and membranes in tapping mode. The colloidal solution zeta potential was measured using a Zetasizer Nano ZS 90. FTIR spectra of the VMT membrane was performed using a Bruker VERTEX 33 unit with the wavenumber range of 400–4000 cm$^{-1}$. The chemistry property of the VMT membrane was analyzed by X-ray photoelectron spectroscopy (XPS) using a Thermo Fisher ESCALAB Xi$^{+}$ instrument with monochromated Al-Kα radiation.

### Electrical measurements

The ion transport properties and osmotic energy conversion properties of the VMT-PVDF heterogeneous membrane were measured using a Keithley 2450 (Keithley Instruments) sourcemeter. The VMT-PVDF heterogeneous membrane was mounted between two compartments of the cell. Electrolyte solutions with high concentration and low concentration were added respectively in the two half-cells, and transmembrane currents were measured using a pair of homemade Ag / AgCl electrodes. The I−V curves were measured in the presence of a salinity gradient by applying the scanning voltage of the from −1 V to + 1 V with the step voltage of 0.01 V. The V$_{OC}$ (interception at zero current) and I$_{SC}$ (interception at zero voltage) could be obtained from the I−V curves, respectively. To eliminate the imbalanced redox potential between the electrode and the salt solutions, agarose salt bridges were employed. For practical osmotic energy conversion measurements, the VMT-PVDF heterogeneous membrane was connected to an external circuit through the electrodes in the reservoirs to supply an electrical load resistor. All electrochemical tests were performed at room temperature and each measurement was repeated at least three times to ensure accuracy and reliability of results.

The cation transfer number ($t_+$) and energy conversion efficiency ($\eta_{max}$) were calculated by the following equation:

$$t_+ = \frac{1}{2}\left( \frac{E_{diff}}{\frac{RT}{zF}\ln\left(\frac{\gamma_{c_H} C_H}{\gamma_{c_L} C_L}\right)} + 1 \right) \tag{3}$$

$$\eta_{max} = \frac{1}{2}(2t_+ - 1)^2 \tag{4}$$

where $R$, $T$, $z$, $F$, $\gamma$, $C_H$ and $C_L$ referred to the universal gas constant, temperature, ion charge valence, Faraday constant, activity coefficient and high and low salt concentrations, respectively. $E_{diff}$ was measured under different concentration gradients.

## Numerical simulation

Numerical simulation was performed using a commercial finite-element software package COMSOL (version 4.3) Multiphysics. The Poisson–Nernst–Planck (PNP) equations which were employed to quantitively describe the ionic mass transport process as depicted below:

$$\vec{j_i} = D_i \left( \nabla c_i + \frac{z_i F c_i}{RT} \nabla \varphi \right) \tag{5}$$

$$\nabla^2 \varphi = -\frac{F}{\varepsilon} \sum_i z_i c_i \tag{6}$$

$$\nabla \cdot \vec{j_i} = 0 \tag{7}$$

where, $j_i$, $D_i$, $c_i$, $z_i$, $\varphi$, and $\varepsilon$ were the ionic flux, diffusion coefficient, ion concentration, valence number for each species $i$, electrical potential, and dielectric constant of the electrolyte solution, respectively. $F$, $R$, and $T$ were the Faraday constant, universal gas constant, and absolute temperature, respectively. The simulation system contained two electrolyte reservoirs connected by a two-segment nanochannel composed of the VMT and the PVDF parts (Supplementary Fig. 21). The external voltage was applied on the boundary $W_1$ and the wall $W_2$ offered reference potential. The ion flux had zero normal components at boundaries:

$$\vec{n} \cdot \vec{j_i} = 0 \tag{8}$$

The boundary condition for the potential $\varphi$ on the channel walls was:

$$-\vec{n} \cdot \nabla \varphi = \frac{\sigma}{\varepsilon} \tag{9}$$

where, $\sigma$ represented the surface charge density. The surface charge density and other structural parameters were detailed in Supplementary Table 1.

## Ab-initio molecular dynamics (AIMD) calculation

Ab-initio Born–Oppenheimer MD simulation was conducted to explore the structure and dynamics of hydrated ions in VMT nanochannel using the CP2K package[44]. The hybrid Gaussian and plane waves (GPW) was used that the electronic density was expanded using plane waves and the cutoff is 500 Ry[45]. The molecularly optimized Gaussian basis sets were used (mDZVP)[46]. The core electrons were treated by Goedecker–Teter–Hutter (GTH) pseudopotentials[47]. PBE+D3 were used for the exchange and correlation functional[48,49]. We study a model system as previous work[30] that 1/4 of the Si atoms in the basal plane of the VMT surface are substituted with Al atoms. It led to the net surface charge of −2e per unit cell, which was the typical structure found in the natural VMT[50]. We used an orthorhombic supercell with the size of 10.79 × 9.35 × 15.30 Å that contained one-layer VMT and confined water solution (Supplementary Fig. 29). The interlayer spacing of nanochannel was same as experimental results. The system contained 19 (19, 21) water molecules and 4 Li$^+$ (4 K$^+$, 2 Mg$^{2+}$) that ensured the charge neutralization. During AIMD, the position of octahedral Mg atoms was fixed to avoid the migration of VMT sheets. The initial structures were from classical MD simulation using the ClayFF force field[51]. Simulations were carried out using the Nose-Hoover thermostat with the damping constant of 100 fs. To reproduce the structure of water at ambient conditions, the higher temperature of 390 K was used to mimic the nuclear quantum effects (NQEs) implicitly that were important to account for the quantum nature of light hydrogen atoms in the simulation of liquid water[52]. Our previous work found PBE+D3 could well capture the structure of water and hydration ions at this temperature[53]. The time step was 1.0 fs. The simulation time was 250 ps, which ensured us to observe the diffusion of ion (excepting for Mg$^{2+}$ because the diffusion was very slow) and water in VMT nanochannel. The former trajectories of 30 ps was discarded for analysis. The self-diffusion coefficient ($D$) of ions was calculated using the Einstein relation that the mean squared displacement (MSD) were extracted from the trajectories, $D = <|\mathbf{r}(t) - \mathbf{r}(0)|^2>/4t$, where $\mathbf{r}(t)$ was the instantaneous position of ions in the x–y plane at time $t$, $<...>$ meant the ensemble average.

## Classical molecular dynamics calculation

The classical MD simulations were performed by using the large-scale atomic/molecular massively parallel simulator (LAMMPS)[54]. The atomic structures were illustrated in Supplementary Fig. 16 where water was confined in VMT sheets with the interlayer spacing of d = 1.53, 2.03 or 2.53 nm. Periodic boundary conditions were used in all the directions with lateral dimensions of 11.41 nm (x), 2.82 nm (y) and 3.06 (4.06 and 5.06) nm (z). The length of the nanochannel was 5.15 nm (in x direction) that was long enough to simulate the distribution of ions in a µm-long channel approximately. The ClayFF force field was used for the system, which had been widely used in atomistic computational modeling for numerous geoscience and materials science applications[51]. The van der Waals interactions were given by the 12-6 Lennard–Jones (L–J) potential. The interatomic parameters were determined by the Lorentz–Berthelot mixing rules. The atoms in the VMT layers were fixed during the simulations. The bond length and angle of water were restrained by the SHAKE algorithm. The long-range Coulombic interactions were computed by the particle–particle particle–mesh (PPPM) algorithm[55]. The concentration of the anion was kept at 2.66 mol kg$^{-1}$ water. The number of the cation was determined by the role of charge neutralization. The quantity of water molecules in the nanochannel was determined by equilibrating the system under the pressure criterion (|px| <50 MPa and |py| <50 MPa). Simulations were carried out at 300 K using the Nose-Hoover thermostat with a damping constant of 1 ps. The time step was 1.0 fs. The systems were equilibrated in the NVT ensemble for 30 ns and the last 25 ns was used for analysis.

## Data availability

The data supporting the findings of this study are available within the article and its Supplementary Information files and are available from the corresponding authors upon request. All data generated in this study are provided in the Supplementary Information/Source Data file. Source data are provided with this paper.

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

## Acknowledgements

This work was supported by National Key R&D Program of China (Nos. 2022YFC2904302, L. W.), the Key R&D program of Shaanxi Province (Nos. 2022SF-134, J. W.) and the National Natural Science Foundation of China (Nos. 12102324, K. Z.). We express our gratitude to Dr. Fang Song from the Instrument Analysis Center of Xi'an University of Architecture and Technology for providing valuable support in the TEM analysis. We also appreciate the assistance provided by the Computing Center in Xi'an, Xi'an University of Architecture and Technology Branch Center.

## Author contributions

J.W. and L.W. designed the experiments. J.W., Z.C., S.Z.L. D.X.H. and Y.F. performed the membrane fabrication and characterization experiments. J.W. and K.Z. performed the molecular dynamics simulations. J.W and Z.C. wrote the paper. X.D.W., M.L.H, Y.F., Y.Z.L and Z.Y.S. contributed to the project discussions and manuscript writing.

## Competing interests

The authors declare no competing interests.
