## [Peer Review File · Nature Communications]

Unlocking Osmotic Energy Harvesting Potential in Challenging Real-World Hypersaline Environments through Vermiculite-Based Hetero-NanochannelsREVIEWER COMMENTS

Reviewer #1 (Remarks to the Author):

This manuscript presents the fabrication of highly cation-selective 2D VMT membranes by depositing ultra-thin VMT laminates on a porous membrane support. The authors systematically investigated the structural morphology, surface properties, ion transport properties, and osmotic energy harvesting performance of the prepared VMT membranes. The authors also demonstrated a high power density of 25.9 W/m², surpassing most reverse electrodialysis membrane systems reported so far. It could, therefore, be a valuable contribution. However, for publication, the following points should be considered:

- Line 117: TEM images should have higher resolution, clearly displaying wrinkles and layers.
- Line 119: The SEM image does not clearly show the lateral dimensions of VMT. Provide a clearer SEM image.
- Line 126: Include a top-view SEM image after VMT deposition, as the SEM image of the PVDF porous membrane alone does not provide useful information.
- Line 145: Provide a better AFM image with good contrast for Fig 1f, showing VMT layers clearly.
- Line 202: Clarify why the VMT membrane in this work shows a higher osmotic power density than other membranes with similar or higher charge density, even though the -50 mV surface charge density of VMT is not significantly lower than -42 mV of GO and -40 mV of Ti₃C₂T_x Mxene nanosheet. Additionally, several papers have reported the surface charge density of BN within the range of -50 mV to -80 mV. Also, convert zeta potential values to surface charge density for a more informative comparison.
- Line 242: Elaborate on the relationship between interlayer spacing, Debye length, and ionic concentration.
- The supplementary SEM image shows a substantial amount of chloride ions. However, the measured cation transference number (line 317) is close to 1, which contradicts the SEM image. Provide the ratio of Na⁺ counts to Cl⁻ counts in the SEM/EDS data for clarification.
- Line 465: The authors stated that monovalent ions such as Li⁺ and K⁺ exhibited strong adsorption on the surface of VMT compared to Mg²⁺. Additionally, the MSD curve showed the order of diffusion: K⁺ > Li⁺ > Mg²⁺. However, in figure 4e, the power density with LiCl is close to that with MgCl₂. The authors should provide a more comprehensive explanation for this result.

Reviewer #2 (Remarks to the Author):

The study focusses on heterogenous nanofluidic system by utilizing a scalable method (VAF) using large sizes (10-12 μm) nanoflakes. They proposed a dual-separation mechanism (when the higher concentration gradient was on pvdf support side specifically) which helped in achievement of power density of 33.76 W/m² using osmotic conversion which was unusually higher than the reported studies.

All the results and performances of system are well backed by simulation and counter experiments. The study can be accepted but it needs further improvement because of very generalized methodology and weak introduction.

Introduction needs to be revised a bit with respect to justification for utilizing vermiculite. Why was vermiculite used in this study. What was the hypothesis. Novelty should be clearly mentioned.

The vacuum assisted filtration technique is not commonly considered as "SCALABLE method" while the authors claimed this as a scalable technique. Any comments?

Materials and Methods can be improved since the steps related to 2D VMT synthesis and membrane fabrication are not clear. The following details can help the viewers to clearly understand the methodology.

- The mass ratio of VMT NaCl used using reflux apparatus for Na-VMT synthesis
- The mass ratio of Na-VMT with LiCl for synthesis of Li-VMT
- How much H₂O₂ was added as a delaminating agent?
- How was the large sized VMT flakes were separated from the bulk solution? Please mention the centrifugation speeds. Also mention the detailed washing steps for finalized VMT.
- How were the flakes dried?
- How the VMT suspension was made. Was it casual mixing and shaking or sonication was also used to RE-DISPERSE the VMT powder in the aqueous suspension?
- How much quantity of VMT aqueous suspension was used to synthesize membrane with different thicknesses. What was the VMT loading (mg / cm²) for different membranes?

Here are some comments for further improvement of manuscript.

- Line 131-132: Does that means that membrane has brittle properties?
- Lines 134-137: where they mentioned the VMT membrane's chemical resistance under challenging conditions after immersion in base or acid for 7 days: Has the chemical resistance of the VMT membrane been tested under varied temperatures, and if so, how does the membrane performance vary?
- Line 147-149: What was the reason for increased surface roughness and wrinkles when thickness was reduced?
- Line 180-182: Is it the proposed novelty of study?
- Line 572-573: The equation mentioned in manuscript should be properly titled (Name of group or study etc)

Review #1

General comment

This manuscript presents the fabrication of highly cation-selective 2D VMT membranes by depositing ultra-thin VMT laminates on a porous membrane support. The authors systematically investigated the structural morphology, surface properties, ion transport properties, and osmotic energy harvesting performance of the prepared VMT membranes. The authors also demonstrated a high power density of 25.9 W/m², surpassing most reverse electrodialysis membrane systems reported so far. It could, therefore, be a valuable contribution. However, for publication, the following points should be considered:

Response to general comment

We thank the reviewer very much for the positive evaluation of our work. All the revised text can be found in the manuscript with changes marked.

Comment 1

Line 117: TEM images should have higher resolution, clearly displaying wrinkles and layers.

Response to comment 1

We appreciate the reviewer's kind comments and apologize for the confusion caused by the unclear TEM images. As suggested by the reviewer, we have replaced the original image with a clearer one to depict the morphology of the monolayer VMT nanosheets in the revised Supplementary Information as follows:

Supplementary Figure 4. TEM images of VMT nanosheet and EDS mappings of oxygen and silicon on the VMT nanosheet. Scale bar, 500 nm.

Note on revised Supplementary Figure 4:

In the initial revision, we identified a mistake in the submitted TEM image, which did not accurately represent the prepared vermiculite nanosheets. We kindly request your permission to correct this error by including an accurate TEM image of the prepared vermiculite nanosheets in Supplementary Figure 4. Additionally, we are providing corresponding energy dispersive spectroscopy (EDS) results with elemental distributions to strongly confirm the authenticity of the TEM images. We sincerely apologize for this oversight and any unintended extension of time it may have caused.

Comment 2

- Line 119: The SEM image does not clearly show the lateral dimensions of VMT. Provide a clearer SEM image.

Response to comment 2

We thank the reviewer for the kind suggestion. We acknowledge the limitation of the original SEM image in depicting the lateral dimensions of VMT nanosheets accurately. We have incorporated additional SEM images with different magnifications that facilitate a clear evaluation of the lateral size of the VMT nanosheets in the revised Supplementary Information as follows:

Supplementary Figure 5. SEM image of the VMT nanosheets with different magnifications. Scale bar, 1 μm . **b, Scale bar, 2 μm .**

Comment 3

- Line 126: Include a top-view SEM image after VMT deposition, as the SEM image of the PVDF porous membrane alone does not provide useful information.

Response to comment 3

We thank the reviewer for the insightful comment. In order to facilitate the readers to have a clearer understanding of the formation process of the 2D VMT membrane, we have included a top-view SEM image of the membrane after the deposition of VMT nanosheets on the PVDF substrate in the revised manuscript (Lines 146-151, Page 8) as follows:

Supplementary Figure 7. Surface SEM images of the membranes. a, Surface SEM image of the PVDF porous substrate. Scale bar, 10 μm . b, Top-view SEM image after VMT nanosheets deposition. Scale bar, 2 μm

“The VMT nanochannel membrane was prepared by parallelly stacking the as-obtained VMT nanosheets on polyvinylidene fluoride (PVDF) porous substrate using a vacuum-assisted filtration method (Supplementary Figs. 6 and 7). The porous surface of PVDF was entirely covered by the deposition of VMT nanosheets, with no apparent voids or defects on the surface of the VMT layer.”

Comment 4

- Line 145: Provide a better AFM image with good contrast for Fig 1f, showing VMT layers clearly.

Response to comment 4

We thank the reviewer for the kind reminder. As suggested by the reviewer, we prepared thicker membranes to enhance the contrast for clearer membrane morphology, and incorporated a better AFM image into the revised manuscript as follows:

Fig. 1 Fabrication and characterization of the VMT nanosheets and VMT membrane. **a**, Schematic illustration of the preparation of the large-scale VMT nanosheets. **b**, AFM image and height profile of the nanosheets corresponding to the AFM image. Scale bar, 2 μm . **c**, Optical microscopy image of the large-scale VMT nanosheets. Scale bar, 20 μm . **d**, SEM image of the cross-sectional morphology of the VMT membrane. Scale bar, 1 μm . **e**, Stress-strain curve of VMT membrane. **f**, AFM image of the VMT membrane and height profile of the VMT membrane corresponding to the AFM image. Scale bar, 1 μm . **g**, Optical image of a VMT membrane.

Comment 5

- Line 202: Clarify why the VMT membrane in this work shows a higher osmotic power density than other membranes with similar or higher charge density, even though the -50 mV surface charge density of VMT is not significantly lower than -42 mV of GO and -40 mV of Ti_3C_2Tx Mxene nanosheet. Additionally, several papers have reported the surface charge density of BN within the range of -50 mV to -80 mV. Also, convert zeta potential values to surface charge density for a more informative comparison.

Response to comment 5

We thank the reviewer for the kind comment. The ion selectivity and permeability are the two most critical parameters of nanofluidic membranes that determine osmotic energy conversion performance. The ion selectivity is notably influenced by the surface charges of the nanochannels, as these charges have a direct impact on the ion distribution within confined channels. While the zeta potential value of VMT is not significantly higher than that of GO and Ti_3C_2Tx nanosheets, we have confirmed the excellent ion selectivity of the VMT membrane through both experimental and theoretical investigations. To provide a more detailed comparison, we calculated the membrane surface charge density from the zeta potential value using the Gouy-Chapman equation as follows (*J. Phys. Chem. Lett.* 2012, 3.7, 867-872):

$$\sigma_s = \frac{2\varepsilon k T \kappa}{ze} \sinh\left(\frac{ze\zeta}{2kT}\right) \quad (1)$$

where ε , z , and e refer to the dielectric constant, valence of the counterions, and elementary charge, respectively and κ represents the reciprocal of the Debye length, which is given by the expression $\sqrt{I}/0.304$. Where I is the ionic strength defined as $(1/2)\sum z_i^2 [x_i]$, and x_i is the molar concentration of the i species, and z_i is ionic valence. The VMT nanosheets, when dispersed in a 0.01 M KCl solution at pH=7, exhibited a stabilized zeta potential of -50 mV, with a calculated surface charge density of -13.4 mC/m².

Furthermore, the high ion permeability is also a pivotal factor contributing to the exceptional energy harvesting efficiency of the VMT membrane. Specifically, due to the micrometer-scale lateral dimensions of VMT nanosheets prepared in our study, pinholes and other structural defects in membranes originating from unideal stacking of nanosheets can be efficiently avoided even with small deposition amounts. Benefiting from this, ultrathin VMT membrane with the thickness on the scale of tens of

nanometers was proposed for RED application, which is much lower than the reported 2D membranes built with other nanosheet such as GO and Mxene. (*ACS Nano* 2019, 13(8), 8917-8925; *Adv. Sci.* 2020, 7(12), 2000286) Given that ion permeability is in inverse proportion to the length of the ion transporting path through the nanochannel, the ultrathin VMT membrane imposes lower resistance on ion transporting and thereby achieves high permeability.

As kindly suggested by the reviewer, BN is a promising material for osmotic energy applications due to its high surface charge density (*Nature* 2013, 494, 455–458; *Joule* 2020, 4(1), 247-261; *Nano Lett.* 2021, 21, 10, 4152–4159). However, based on our literature investigation, synthesis of high-quality BN nanosheets remains challenging in practical applications. For instance, the synthesis process of large bulk BN crystals typically requires high-temperature and high-pressure conditions (*Nat. Mater.* 2004, 3, 404), posing a challenge for achieving cost-effective practical use. Furthermore, the limited size and thickness of flakes of these BN crystals generated through mechanical exfoliation present inherent obstacles to achieving reliable and repeatable production (*J. Am. Chem. Soc.* 2017, 139(18), 6314-6320). In comparison, the atomic-thick VMT nanosheets with large lateral size could be facily obtained via milder eco-friendly liquid exfoliation method, the simple preparation process and low cost of raw VMT make it more favorable for practical applications. To provide a comprehensive understanding of the novelty underpinning our current work, we have enriched the introduction section by elucidating the rationale behind selecting 2D VMT as the building material for constructing membranes.

It should be noted that utilizing a single membrane material to propel osmotic energy harvesting from laboratory research to practical industrial applications poses a considerable challenge. Lei et al. successfully engineered a hybrid membrane, combining Ti_3C_2Tx and BN, demonstrating exceptional stability and reduced internal resistance, thereby enhancing the efficiency of salinity gradient energy harvesting. (*ACS Nano* 2021, 15, 6594-6603) Additionally, recent studies have shown that the integration of one-dimensional (1D) materials into 2D lamellar membranes can significantly boost power density while enhancing mechanical strength (*Nat. Commun.* 2019, 10, 2920). Building upon the promising osmotic energy harvesting performance demonstrated in this study using VMT membranes, our future research endeavors will focus on combining VMT with other low-dimensional nanomaterials, including BN

nanosheet, to leverage the inherent advantages of diverse materials fully, thereby further enhance energy harvesting efficiency.

A detailed description has been added in the revised manuscript (Lines 52-72, Pages 3-4; Lines 201-210, Page 10) as follows:

“2D Nanofluidic channel systems with controllable ion transport have emerged as a novel approach for osmotic energy harvesting⁷⁻⁹. For instance, a single-layer MoS₂ has achieved a power density of approximately 10⁶ Wm⁻², which is a million times higher than that attained by traditional exchange membranes⁶. ~~While the large scale application of such devices under realistic operating conditions remains a challenge, these results certainly provide a new source of inspiration for the development of nanochannel-based osmotic power harvesting.~~ Although these results certainly provide a new source of inspiration for the development of osmotic power harvesting technology, there are many challenges currently still impede the industrial applications of 2D nanochannel membranes. For instance, the exfoliation of most 2D nanomaterials typically requires the use of harsh chemicals such as strong acids, oxidants, or solvents to break the out-of-plane bonds in the layered structure^{10,11}. In addition, the process of synthesizing precursors, such as MAX phase for MXene, that do not occur naturally, can be intricate¹². Vermiculite (VMT) is a cost-effective naturally occurring clay mineral with a global production of approximately 500,000 tons¹³. Because Si⁴⁺ and Al³⁺ are uniformly substituted with low-valent ions, VMT layers naturally exhibit negative charge property. These charges can be balanced by the adsorption of cations, such as Mg²⁺ and K⁺, between the layers. Preliminary investigations have demonstrated that bulk VMT can be readily exfoliated in aqueous solution to obtain two-dimensional nanosheets with atomic-level thickness by the more environmentally friendly and gentler ion-exchange method^{14,15}.

~~It worth noting that, although 2D lamellar membranes show advantages in large scale preparation relative to nanopores fabricated through intricate techniques such as ion beam irradiation and ion tracking etching, there are many challenges currently still impede the industrial scalability of 2D nanochannel membranes. For instance, the exfoliation of most 2D nanomaterials typically requires the use of harsh chemicals such as strong acids, oxidants, or solvents to break the out of plane bonds in the layered structure. In addition, the synthesis of precursors, such as MAX phase for MXene, that do not occur naturally, can be intricate. Given the above findings, the facile synthesis method for producing VMT nanosheets and nanochannels render VMT membrane a~~

~~promising choice for large-scale manufacturing and practical osmotic power generation (Fig. 1g and Supplementary Fig. 14).~~

The surface charge density (σ_s) of VMT nanosheets was also calculated from the zeta potential value using the Gouy-Chapman equation as follows²⁷:

$$\sigma_s = \frac{2\varepsilon k T \kappa}{ze} \sinh\left(\frac{ze\zeta}{2kT}\right) \quad (1)$$

where ε , z and e refer to the dielectric constant, valence of the counterions, and elementary charge, respectively and κ represents the reciprocal of the Debye length, which is given by the expression $\sqrt{I}/0.304$. Where I is the ionic strength defined as $(1/2)\sum z_i^2 [x_i]$, and x_i is the molar concentration of the i species, and z_i is ionic valence. The VMT nanosheets, when dispersed in a 0.01 M KCl solution at pH=7, exhibited a stabilized zeta potential of -50 mV, with a calculated surface charge density of -13.4 mC/m².

Ref 10: Yang, Q. *et al.* Ultrathin graphene-based membrane with precise molecular sieving and ultrafast solvent permeation. *Nat. Mater.* **16**, 1198-1202 (2017).

Ref 11: Wang, Z. & Mi, B. Environmental applications of 2D molybdenum disulfide (MoS₂) nanosheets. *Environ. Sci. Technol.* **51**, 8229-8244 (2017).

Ref 12: Alhabeab, M. *et al.* Guidelines for synthesis and processing of two-dimensional titanium carbide (Ti₃C₂T_x MXene). *Chem. Mater.* **29**, 7633-7644 (2017).

Ref 13: Xia, Z. *et al.* Tunable Ion Transport with Freestanding Vermiculite Membranes. *ACS Nano* **16**, 18266-18273 (2022).

Ref 14: Shao, J.-J., Raidongia, K., Koltonow, A. R. & Huang, J. Self-assembled two-dimensional nanofluidic proton channels with high thermal stability. *Nat. Commun.* **6**, 7602 (2015).

Ref 15: Zhang, T. *et al.* Precise cation recognition in two-dimensional nanofluidic channels of clay membranes imparted from intrinsic selectivity of clays. *ACS Nano* **16**, 4930-4939 (2022).

Ref 27: Konkena, B. & Vasudevan, S. Understanding aqueous dispersibility of graphene oxide and reduced graphene oxide through pKa measurements. *J. Phys. Chem. Lett.* **3**, 867-872 (2012).”

Comment 6

- Line 242: Elaborate on the relationship between interlayer spacing, Debye length, and ionic concentration.

Response to comment 6

We thank the reviewer for the kind comment. According to the Electric Double Layer (EDL) theory, the Debye length represents the approximate distance over which the potential at a charged surface dissipates into the bulk solution. This length is closely linked to electrolyte concentration and valence and can be theoretically calculated using the equation (*Angew. Chem. Int. Ed.* 2023, 62, e202303582):

$$\lambda_D = \sqrt{\frac{\varepsilon RT}{\sum_{i=1}^N F^2 Z_i^2 C_{i,0}}} \quad (2)$$

Here, ε is the solution permittivity, R , and T are the gas constant and temperature, respectively; F is the Faraday constant; N is the total number of ionic species; Z_i and $C_{i,0}$ are the valence and bulk concentration of i th ionic species, respectively. With increasing ion concentration, the Debye length decreases. In a 0.1 M KCl solution, the Debye length is approximately 1 nm, whereas in a 1 M KCl solution, it reduces to about 0.3 nm.

The thickness (h) of a monolayer VMT nanosheet is defined by the equation (*Extreme Mech. Lett.* 2022, 57, 101921): $h = d_{T-B} + r_T + r_B$, where d_{T-B} is the sum of the vertical distance between the topmost and bottommost atoms, r_T and r_B is the van der Waals radius of the topmost and bottommost atoms, respectively. According to DFT calculations of structure relaxation, h is calculated to be 9.6 Å. The X-ray diffraction pattern (XRD) in Supplementary Figure 8 reveals a d-spacing of 1.53 nm, allowing us to estimate the diameter of the interlayer nanochannel as 0.6 nm.

Based on above analysis, when the KCl concentration falls below 0.1 M, the EDLs on opposite surfaces overlap within the nanochannels. Consequently, the ion distribution and population become significantly influenced by the surface charges, owing to the robust electrostatic interaction. As a result, the negatively charged VMT nanochannels demonstrate exceptional cation selectivity. These findings are consistent with our original manuscript (Fig. 2b), wherein a deviation from bulk ionic conductance was observed when the electrolyte concentration dropped below 0.1 M. Such transport phenomenon aligns with the typical surface-charge-governed ion diffusion behavior, as observed in numerous 2D nanofluidic systems (*Nano Energy* 2020, 76, 105113; *Angew. Chem. Int. Ed.* 2023, 62,19).

It is crucial to emphasize that the Debye length only provided a qualitative depiction of the spatial extent of the electrostatic effects within the EDL. However, in conventional Poisson-Boltzmann theory, ions in the EDL are often treated as point charges (*Anal.*

Chim. Acta 2019, 1059, 68–79). While such simplification may be acceptable for microscale channels, a more detailed consideration of ion size and interactions between ions become essential in the highly confined channels.

A detailed description has been added in the revised manuscript (Lines 252-288, Pages 11-13) as follows:

“Based on the XRD results, the interlayer spacing of VMT membrane was approximately 1.53 nm. The thickness of VMT (h) is defined by geometric thickness (Supplementary Fig.16), $h = d_{T-B} + r_T + r_B$, where d_{T-B} is the sum of the vertical distance between the topmost and bottommost atoms, r_T and r_B is the van der Waals radius of the topmost and bottommost atoms, respectively³¹. The h can be obtained by performing the DFT calculation of structure relaxation that $h = 9.6 \text{ \AA}$. Based on this, the height of the nanochannel can be estimated to be 0.6 nm. According to the EDL theory, the Debye length represents the approximate distance over which the potential at a charged surface dissipates into the bulk solution. This length is closely linked to electrolyte concentration and valence and can be theoretically calculated using the equation³²:

$$\lambda_D = \sqrt{\frac{\varepsilon RT}{\sum_{i=1}^N F^2 Z_i^2 C_{i,0}}} \quad (2)$$

Here, ε is the solution permittivity, R , and T are the gas constant and temperature, respectively; F is the Faraday constant; N is the total number of ionic species; Z_i and $C_{i,0}$ are the valence and bulk concentration of i th ionic species, respectively. It could be observed that with increasing ion concentration, the Debye length decreases. For example, in a 0.1 M KCl solution, the Debye length is approximately 1 nm, whereas in a 1 M KCl solution, it reduces to about 0.3 nm. Based on calculation of VMT channel diameter, it could be inferred that when the KCl concentration falls below 0.1 M, the EDLs on opposite surfaces overlap within the nanochannels. Consequently, the ion distribution and population become significantly influenced by the surface charges, owing to the robust electrostatic interaction. As a result, at high concentrations, the corresponding transmembrane ionic conductance demonstrated a linear correlation with solution concentration, similar to bulk behavior. However, a deviation from bulk ionic conductance was observed when the electrolyte concentration dropped below 0.1 M as shown in Fig. 2c. Such a phenomenon of ionic transport was consistent with the typical surface-charge-governed ion diffusion behavior observed in previous reports, which

was attributed to the fact that ion diffusion was significantly impacted by surface charges when the channel size was reduced to nano-scale^{33,34}. It should be noted that the Debye length provides a qualitative depiction of the spatial extent of the electrostatic effects within the EDL. However, in highly confined channels, factors such as ion size and interactions between ions become crucial. In conventional Poisson-Boltzmann theory, ions within the EDL are commonly treated as point charges, neglecting these intricate details³⁵.

Ref 31: Tian, S. *et al.* Investigation and understanding of the mechanical properties of MXene by high-throughput computations and interpretable machine learning. *Extreme Mech. Lett.* **57**, 101921 (2022).

Ref 32: Chu, C. W., Fauziah, A. R. & Yeh, L. H. Optimizing membranes for osmotic power generation. *Angew. Chem. Int. Ed.* **62**, e202303582 (2023).

Ref 35: Li, J., Peng, R. & Li, D. Effects of ion size, ion valence and pH of electrolyte solutions on EOF velocity in single nanochannels. *Anal. Chim. Acta* **1059**, 68-79 (2019).”

Comment 7

- The supplementary SEM image shows a substantial amount of chloride ions. However, the measured cation transference number (line 317) is close to 1, which contradicts the SEM image. Provide the ratio of Na⁺ counts to Cl⁻ counts in the SEM/EDS data for clarification.

Response to comment 7

We thank the reviewer for the kind comment. Following the reviewer's suggestion, we have provided additional content information corresponding to the mapping images in the supplementary information. These supplementary data revealed a significantly higher amount of K⁺ than Cl⁻ on the VMT membrane surface, confirming the VMT membrane's high cation selectivity. It is crucial to acknowledge that EDS spectroscopy serves as a qualitative analytical tool, supporting the cation selectivity of VMT membranes. To assess the practical cation transference number (t_+) during osmotic energy conversion, we conducted calculations based on equations 3-4 as listed in the original manuscript. Our findings indicate stable cation transference numbers of 0.8, even under high electrolyte concentrations of 1M (Supplementary Fig. 24). A detailed description has been added in the revised manuscript (Lines 305-308, Page 14) as follows:

“The content information corresponding to the mapping images in the Supplementary

Fig. 18 indicates a significantly higher amount of K^+ compared to Cl^- , further confirming the VMT membrane's pronounced cation selectivity.”

Supplementary Figure 18. EDS mappings of potassium and chlorine on VMT-based nanochannel membrane after immersing in 1 M KCl solution for 10 h. The negatively charged VMT surface showed preferential selectivity for K^+ due to strong electrostatic effect. Scale bar, 5 μm .

Comment 8

- Line 465: The authors stated that monovalent ions such as Li^+ and K^+ exhibited strong adsorption on the surface of VMT compared to Mg^{2+} . Additionally, the MSD curve showed the order of diffusion: $K^+ > Li^+ > Mg^{2+}$. However, in figure 4e, the power density with $LiCl$ is close to that with $MgCl_2$. The authors should provide a more comprehensive explanation for this result.

Response to comment 8

We thank the reviewer for their insightful comment. We sincerely apologize for any confusion caused by insufficient explanation. In the context of applying the nanofluidic membrane to osmotic energy harvesting, several factors determine the output power density. Considering the different valences of cations, the ionic current generated by the transport of a specific quantity of divalent Mg^{2+} through the nanochannels equals the ionic current produced by twice the amount of Li^+ passing through these channels. As a result, our AIMD simulation revealed a higher transmembrane permeation rate for

Li⁺ compared to Mg²⁺, however, the power density in the MgCl₂ solution system is comparable to that in the LiCl solution system. This observation aligns with previous studies investigating the impact of cation type on output power density (*Angew. Chem. Int. Ed.* 2022, 61(41), e202206152; *Nat. Commun.* 2020, 11(1), 875). A detailed description has been added in the revised manuscript (Lines 549-557, Pages 24-25) as follows:

“It should be noted that considering the different valences of cations, the ionic current generated by the transport of a specific quantity of divalent Mg²⁺ through the nanochannels equals the ionic current produced by twice the amount of Li⁺ passing through these channels. As a result, our AIMD simulation revealed a higher transmembrane permeation rate for Li⁺ compared to Mg²⁺, however, the power density in the MgCl₂ solution system is comparable to that in the LiCl solution system. This observation aligns with previous studies investigating the impact of cation type on output power density^{18,42}.

Ref 18: Ding, L. *et al.* Bioinspired Ti₃C₂Tx MXene-Based Ionic Diode Membrane for High-Efficient Osmotic Energy Conversion. *Angew. Chem. Int. Ed.* **61**, e202206152 (2022).

Ref 42: Zhang, Z. *et al.* Improved osmotic energy conversion in heterogeneous membrane boosted by three-dimensional hydrogel interface. *Nat. Commun.* **11**, 875 (2020).”

Review #2

General comment

The study focusses on heterogenous nanofluidic system by utilizing a scalable method (VAF) using large sizes (10-12 μm) nanoflakes. They proposed a dual-separation mechanism (when the higher concentration gradient was on pvdf support side specifically) which helped in achievement of power density of 33.76 W/m^2 using osmotic conversion which was unusually higher than the reported studies. All the results and performances of system are well backed by simulation and counter experiments. The study can be accepted but it needs further improvement because of very generalized methodology and weak introduction.

Response to general comment

We greatly appreciate the reviewers for their detailed and constructive suggestions. We have tried our best to comply with the suggestions in the revised manuscript. Point-by-point responses to the comments are enclosed.

Comment 1

Introduction needs to be revised a bit with respect to justification for utilizing vermiculite. Why was vermiculite used in this study. What was the hypothesis. Novelty should be clearly mentioned.

Response to comment 1

We extend our appreciation to the reviewer for their valuable feedback. In accordance with the reviewer's suggestion, we have enriched the introduction section by elucidating the rationale behind selecting 2D vermiculite as the primary building material for constructing laminated membranes. This addition aims to provide a comprehensive understanding of the novelty underpinning our current work. Further details can be found in the revised manuscript (Lines 52-72, Pages 3-4) as follows:

“2D Nanofluidic channel systems with controllable ion transport have emerged as a novel approach for osmotic energy harvesting⁷⁻⁹. For instance, a single-layer MoS_2 has achieved a power density of approximately 10^6 Wm^{-2} , which is a million times higher

than that attained by traditional exchange membranes⁶. ~~While the large-scale application of such devices under realistic operating conditions remains a challenge, these results certainly provide a new source of inspiration for the development of nanochannel-based osmotic power harvesting.~~ Although these results certainly provide a new source of inspiration for the development of osmotic power harvesting technology, there are many challenges currently still impede the industrial applications of 2D nanochannel membranes. For instance, the exfoliation of most 2D nanomaterials typically requires the use of harsh chemicals such as strong acids, oxidants, or solvents to break the out-of-plane bonds in the layered structure^{10,11}. In addition, the process of synthesizing precursors, such as MAX phase for MXene, that do not occur naturally, can be intricate¹². Vermiculite (VMT) is a cost-effective naturally occurring clay mineral with a global production of approximately 500,000 tons¹³. Because Si⁴⁺ and Al³⁺ are uniformly substituted with low-valent ions, VMT layers naturally exhibit negative charge property. These charges can be balanced by the adsorption of cations, such as Mg²⁺ and K⁺, between the layers. Preliminary investigations have demonstrated that bulk VMT can be readily exfoliated in aqueous solution to obtain two-dimensional nanosheets with atomic-level thickness by the more environmentally friendly and gentler ion-exchange method^{14,15}.

~~It worth noting that, although 2D lamellar membranes show advantages in large-scale preparation relative to nanopores fabricated through intricate techniques such as ion beam irradiation and ion tracking etching, there are many challenges currently still impede the industrial scalability of 2D nanochannel membranes. For instance, the exfoliation of most 2D nanomaterials typically requires the use of harsh chemicals such as strong acids, oxidants, or solvents to break the out-of-plane bonds in the layered structure. In addition, the synthesis of precursors, such as MAX phase for MXene, that do not occur naturally, can be intricate. Given the above findings, the facile synthesis method for producing VMT nanosheets and nanochannels render VMT membrane a promising choice for large-scale manufacturing and practical osmotic power generation (Fig. 1g and Supplementary Fig. 14).~~

Ref 10: Yang, Q. *et al.* Ultrathin graphene-based membrane with precise molecular sieving and ultrafast solvent permeation. *Nat. Mater.* **16**, 1198-1202 (2017).

Ref 11: Wang, Z. & Mi, B. Environmental applications of 2D molybdenum disulfide (MoS₂) nanosheets. *Environ. Sci. Technol.* **51**, 8229-8244 (2017).

Ref 12: Alhabeab, M. *et al.* Guidelines for synthesis and processing of two-dimensional titanium carbide ($\text{Ti}_3\text{C}_2\text{Tx}$ MXene). *Chem. Mater.* **29**, 7633-7644 (2017).

Ref 13: Xia, Z. *et al.* Tunable Ion Transport with Freestanding Vermiculite Membranes. *ACS Nano* **16**, 18266-18273 (2022).

Ref 14: Shao, J.-J., Raidongia, K., Koltonow, A. R. & Huang, J. Self-assembled two-dimensional nanofluidic proton channels with high thermal stability. *Nat. Commun.* **6**, 7602 (2015).

Ref 15: Zhang, T. *et al.* Precise cation recognition in two-dimensional nanofluidic channels of clay membranes imparted from intrinsic selectivity of clays. *ACS Nano* **16**, 4930-4939 (2022).”

Comment 2

The vacuum assisted filtration technique is not commonly considered as “SCALABLE method” while the authors claimed this as a scalable technique. Any comments?

Response to comment 2

We thank the reviewer for the kind comment. In recent years, the scientific community has witnessed a surge of interest in two-dimensional layered membranes, particularly in the domains of gas, molecular, and ion separations. This enthusiasm stems not only from the potential for precise adjusting both the geometry and surface properties of nanochannels of 2D membrane, the scalability also aligns with the practical needs of industrial applications (*Angew. Chem. Int. Ed.* 2023, 62,19; *Nat Commun.* 2016, 7, 10891).

In our study, we expanded the membrane area from 12.56 cm² to 78.5 cm² utilizing the common vacuum-assisted filtration technique. However, as highlighted by the reviewer, vacuum-assisted filtration is not inherently scalable due to the low efficiency and time consuming. In response to this insightful comment, we endeavored to prepare the VMT membrane using the spray coating via a home-made device to further demonstrate scalability. As illustrated in Fig. 1g, the 2D VMT membrane, covering an area of 300 cm² and with a thickness of 3 μm, was successfully prepared within 30 minutes. The resulting membrane displayed a uniformly consistent surface, devoid of any discernible structural defects. A detailed description has been added in the revised manuscript (Lines 184-194, Page 9) as follows:

“In our pursuit of membrane scale-up preparation, we sought to leverage vacuum-assisted filtration, a widely adopted method in the realm of 2D membrane preparation^{23,24}, to increase the membrane size by expanding the filtration area from

12.56 cm² to 78.5 cm². Furthermore, given the low efficiency and time consuming of vacuum-assisted filtration, we explored a more efficient spray coating method to prepare a large-scale VMT membrane, providing a clearer illustration of scalability. As evidenced in Supplementary Figs. 14 and 15, the 2D VMT membrane, covering an area of 300 cm² and with a thickness of 3 μm, was successfully prepared within 30 minutes. The resulting membrane displayed a uniformly consistent surface, devoid of any discernible structural defects (Fig. 1g).

Fig. 1 Fabrication and characterization of the VMT nanosheets and VMT membrane. **a**, Schematic illustration of the preparation of the large-scale VMT nanosheets. **b**, AFM image and height profile of the nanosheets corresponding to the AFM image. Scale bar, 2 μm. **c**, Optical microscopy image of the large-scale VMT nanosheets. Scale bar, 20 μm. **d**, SEM image of the cross-sectional morphology of the VMT membrane. Scale bar, 1 μm. **e**, Stress–strain curve of VMT membrane. **f**, AFM image of the VMT membrane and height profile of the VMT membrane corresponding to the AFM image. Scale bar, 1 μm. **g**, Optical image of a VMT membrane.

Supplementary Figure 14. Schematic of home-made spray coating device.

Supplementary Figure 15. Optical images of the VMT-based nanochannel membranes. a, Expansion of membrane area from 12.56 cm² to 78.5 cm² by vacuum-filtration method. **b,** The membrane covering an area of 300 cm² fabricated by spray coating method.

Ref 23: Xu, W. L. *et al.* Self-assembly: a facile way of forming ultrathin, high-performance graphene oxide membranes for water purification. *Nano Lett.* **17**, 2928-2933 (2017).

Ref 24: Sun, P. *et al.* Selective ion penetration of graphene oxide membranes. *ACS Nano* **7**, 428-437 (2013).”

Comment 3

Materials and Methods can be improved since the steps related to 2D VMT synthesis and membrane fabrication are not clear. The following details can help the viewers to clearly understand the methodology.

-Here are some comments for further improvement of manuscript.

- The mass ratio of VMT NaCl used using reflux apparatus for Na-VMT synthesis
- The mass ratio of Na-VMT with LiCl for synthesis of Li-VMT
- How much H₂O₂ was added as a delaminating agent?
- How was the large sized VMT flakes were separated from the bulk solution? Please mention the centrifugation speeds. Also mention the detailed washing steps for finalized VMT.
- How were the flakes dried?
- How the VMT suspension was made. Was it casual mixing and shaking or sonication was also used
- How much quantity of VMT aqueous suspension was used to synthesize membrane with different thicknesses. What was the VMT loading (mg / cm²) for different membranes

Response to comment 3

We thank the reviewer for the valuable comment. We sincerely apologize for any confusion stemming from the limited detail in the Materials and Methods section concerning the preparation of 2D VMT nanosheets and lamellar membrane. As suggested by the reviewer, we have significantly expanded this section to offer a more comprehensive and transparent description of the preparation method and process. A detailed description has been added in the revised manuscript (Lines 600-619, Page 27) as follows:

“Monolayer VMT nanosheets with large lateral size were synthesized via a two-step ion exchange exfoliation process¹⁴. Initially, 3 g of thermally expanded VMT powder was immersed in 200 mL of saturated sodium chloride (NaCl) solution and refluxed at 120°C with continuous stirring for 24 h. The obtained precipitate, designated as Na-VMT, was washed with deionized (DI) water. Next, Na-VMT was dispersed in 200 mL of a 2 M lithium chloride (LiCl) solution, followed by reflux at 120°C for 24 h to ensure complete ion exchange. Thorough rinsing with DI water was conducted to eliminate residual LiCl, yielding the desired Li-VMT product. Subsequently, the Li-VMT product was immersed in 100 ml of a 30% hydrogen peroxide (H₂O₂) solution, which further delaminated it into the VMT nanosheets. High-speed centrifugation (8000 rpm, 30min) was carried out to effectively remove impurities and multilayered nanosheets, and the resulting sediment was redispersed. ~~Low-speed centrifugation was then performed to isolate large-scale VMT flakes.~~ Low-speed centrifugation (2500 rpm, 30 min) was then

performed to collect the stably dispersed large-scale VMT flakes in the upper solution. The final product was a stable light-yellow suspension of monolayer VMT nanosheets with large dimensions, which were further characterized and analyzed. Membranes with a thickness of 1 μm was prepared by depositing 0.8 mg/cm^2 of nanosheets on the substrate and subsequently the obtained membranes were dried overnight under ambient conditions.”

Comment 4

Here are some comments for further improvement of manuscript.

Response to comment 4

We thank the reviewer for the kind comment. Point-by-point responses to the comments are enclosed.

Specific comment 4-1

-Line 131-132: Does that mean that membrane has brittle properties?

Response to specific comment 4-1

We thank the reviewer for the kind comment. As noted by the reviewer, flexibility is a pivotal mechanical attribute for practical applications in osmotic energy conversion. As depicted in our images, the VMT membrane exhibits excellent flexibility and remains structurally intact even after undergoing multiple folds. This good mechanical property provides great convenience for industrial production of membrane modules for practical applications. A detailed description has been added in the revised manuscript (Lines 156-161, Page 8) as follows:

“The VMT membrane exhibited great mechanical properties and flexibility with a tensile strength of approximately 38 MPa and a fracture strain of 2.52% (Fig. 1e). Flexibility is also a pivotal mechanical attribute for practical applications in osmotic energy conversion. As depicted in our images (Supplementary Fig. 11), the VMT membrane exhibits excellent flexibility and remains structurally intact even after undergoing multiple folds.”

Supplementary Figure 11. Images of the VMT-based nanochannel membrane **a**, The VMT-based nanochannel membrane after undergoing multiple folds. **b**, The unfolded VMT-based nanochannel membrane after multiple folding.

Specific comment 4-2

-Lines 134-137: where they mentioned the VMT membrane's chemical resistance under challenging conditions after immersion in base or acid for 7 days: Has the chemical resistance of the VMT membrane been tested under varied temperatures, and if so, how does the membrane performance vary?

Response to specific comment 4-2

We thank the reviewer for the kind comment. As suggested by the reviewer, chemical resistance at high temperatures is crucial for practical osmotic energy applications, especially concerning industrial wastewater. We conducted experiments to analyze the ion transport behavior in VMT-based nanofluidic membranes after immersion in acidic and alkaline solutions at 80°C for 6 hours, respectively. We confirmed that the employed immersion treatment had almost no impact on the surface-charge-governed ion transport behavior in the VMT nanochannels membrane, indicating the chemical resistance of the VMT membrane at high temperatures. Previous studies have also demonstrated that VMT nanochannels can retain their layered structure and maintain their proton conduction functions even after annealing at 500°C in air, showcasing their extraordinary thermal stability (*Nat. Commun.* 2015, 6(1), 7602). In addition, Tian et al. demonstrated that the VMT membrane does not show an obvious change in electrical conductivity after thermal treatment at 200 °C or even 500 °C in air for 6 h (*Inorg.*

Chem. 2023, 62, 14, 5400–5407). A detailed description has been added in the revised manuscript (Lines 315-334, Page 15) as follows:

“Adequate chemical resistance and mechanical stability in harsh environments are crucial properties for membranes in practical osmotic energy harvesting applications. The VMT membrane showed adequate chemical resistance and robust mechanical stability when subjected to challenging environmental conditions, as demonstrated by its structural integrity after immersion in base or acid for 7 days (Supplementary Fig. 19). Chemical resistance at high temperatures is crucial for practical osmotic energy applications, especially concerning industrial wastewater. Previous study demonstrated that VMT nanochannels can retain their layered structure and maintain their proton conduction functions even after annealing at 500°C in air, showcasing their extraordinary thermal stability¹⁴. We also conducted experiments to analyze the ion transport behavior in VMT-based nanofluidic membranes after immersion in acidic and alkaline solutions with a concentration of 0.001M at 80°C for 6 hours, respectively. We confirmed that the employed immersion treatment had almost no impact on the surface-charge-governed ion transport behavior in the VMT nanochannels membrane, indicating the chemical resistance of the VMT membrane at high temperatures (Supplementary Fig. 20).

Supplementary Figure 20. The Chemical resistance of the VMT-based nanochannel membranes in water at 80°C. **a**, Surface morphology of the VMT membrane before and after immersion in water at pH=3.0 and pH=11 for 6 h at 80°C, respectively. **b**, Conductivity of VMT membrane as a function of salt concentration after immersion in water at pH=3.0 and pH=11 for 6 h at 80°C, respectively.

Ref 14: Shao, J.-J., Raidongia, K., Koltonow, A. R. & Huang, J. Self-assembled two-dimensional nanofluidic proton channels with high thermal stability. *Nat. Commun.* **6**, 7602 (2015).”

Specific comment 4-3

-Line 147-149: What was the reason for increased surface roughness and wrinkles when thickness was reduced?

Response to specific comment 4-3

We thank the reviewer for the kind comment. In the case of a thin VMT membrane, the appearance of wrinkles on the membrane surface may be attributed to the unevenness of the porous substrate. As the deposition quantity gradually increased, abundant uniformly stacked VMT nanosheets effectively coated the substrate, resulting in a relatively flat surface morphology. A detailed description has been added in the revised manuscript (Lines 162-172, Pages 8-9) as follows:

“In our study, the VMT membranes with nano-scale thickness were attempt to prepared to achieve a high ion flux. Here, the prepared VMT membrane was transferred to a silicon wafer with a flat surface and the thickness of the membrane was precisely quantified by AFM (Fig. 1f) and the ultrathin VMT membrane show a uniform thickness distribution. ~~Moreover, even though the nanowrinkles increased as the thickness decreasing, the structure of the VMT membrane remained intact and no holes or other defects were visible (Supplementary Fig. 12).~~ In the case of a thin VMT membrane, the appearance of wrinkles on the membrane surface may be attributed to the unevenness of the porous substrate. However, the structure of the VMT membrane remained intact and no holes or other defects were visible. As the quantity of deposited VMT nanosheets increases, they effectively coat the substrate surface, resulting in a flat morphology due to the uniform stacking of nanosheets (Supplementary Fig. 12).”

Specific comment 4-4

-Line 180-182: Is it the proposed novelty of study?

Response to specific comment 4-4

We thank the reviewer for the kind comment. As previously highlighted in **Comment 1**, unlike the exfoliation of other 2D materials that necessitate the use of harsh chemicals to delaminate the bulk layered precursors into individual atomically thin nanosheets, 2D VMT nanosheets with monolayer thickness and large-scale lateral size can be easily obtained in aqueous solutions through the environmentally friendly and milder ion-exchange method. The selection of VMT nanosheets using this green preparation

technique stands as a key innovation in our study. To enhance the reader's understanding of this innovation point, we have incorporated a detailed description of this aspect into the introduction. Moreover, inspired by the reviewer's valuable suggestion, we endeavored to produce the large-scale membrane by employing an efficient spray coating method. As described in the aforementioned **Comment 2**, we successfully produced a VMT membrane covering an area of 300 cm² with an intact structure within a mere 30 minutes.

Specific comment 4-5

-Line 572-573: The equation mentioned in manuscript should be properly titled (Name of group or study etc)

Response to specific comment 4-5

We thank the reviewer for the kind comment. We apologize for the confusion caused by the Improperly titled equation. A detailed description has been added in the revised manuscript (Lines 658-659, Page 29) as follows:

$$(2t_+ - 1) = \frac{E_{diff}}{\frac{RT}{zF} \ln\left(\frac{\gamma_{cH} C_H}{\gamma_{cL} C_L}\right)} \quad (1)$$

$$\eta_{max} = \frac{1}{2} (2t_+ - 1)^2 \quad (2)$$

$$t_+ = \frac{1}{2} \left(\frac{E_{diff}}{\frac{RT}{zF} \ln\left(\frac{\gamma_{cH} C_H}{\gamma_{cL} C_L}\right)} + 1 \right) \quad (3)$$

$$\eta_{max} = \frac{1}{2} (2t_+ - 1)^2 \quad (4)$$

REVIEWERS' COMMENTS

Reviewer #1 (Remarks to the Author):

The authors have responded to all comments reasonably and revised the manuscripts by addressing all my comments. The manuscript has an improved quality, so I recommend it for publication after addressing the following minor comment.

Minor:

Fig. 1d: It is highly recommended that the authors provide a higher-resolution SEM image to more clearly show the layered structure of the VMT nanosheet membrane.

Review #1

General comment

The authors have responded to all comments reasonably and revised the manuscripts by addressing all my comments. The manuscript has an improved quality, so I recommend it for publication after addressing the following minor comment.

Response to general comment

We thank the reviewer very much for the positive evaluation of our work. All the revised text can be found in the manuscript.

Comment 1

Fig. 1d: It is highly recommended that the authors provide a higher-resolution SEM image to more clearly show the layered structure of the VMT nanosheet membrane.

Response to comment 1

We appreciate the reviewer's kind comments and apologize for the confusion caused by the unclear SEM images. As suggested by the reviewer, we have replaced the original image with a clearer one to depict the layered structure of the VMT membrane in the revised manuscript as follows:

Fig. 1 Fabrication and characterization of the VMT nanosheets and VMT membrane. **a**, Schematic illustration of the preparation of the large-scale VMT nanosheets. **b**, AFM image and height profile of the nanosheets corresponding to the AFM image. Scale bar, 2 μm . **c**, Optical microscopy image of the large-scale VMT nanosheets. Scale bar, 20 μm . **d**, SEM image of the cross-sectional morphology of the VMT membrane. Scale bar, 3 μm . **e**, Stress-strain curve of VMT membrane. **f**, AFM image of the VMT membrane and height profile of the VMT membrane corresponding to the AFM image. Scale bar, 1 μm . **g**, Optical image of a VMT membrane.